# PRDX6 augments selenium utilization to limit iron toxicity and ferroptosis

Hiroaki Fujita [1], Yu-ki Tanaka[2,5], Seiryo Ogata[3,5], Noriyuki Suzuki[2], Sota Kuno [1,4], Uladzimir Barayeu[3], Takaaki Akaike [3], Yasumitsu Ogra[2] & Kazuhiro Iwai [1]

Ferroptosis is a form of regulated cell death induced by iron-dependent accumulation of lipid hydroperoxides. Selenoprotein glutathione peroxidase 4 (GPX4) suppresses ferroptosis by detoxifying lipid hydroperoxides via a catalytic selenocysteine (Sec) residue. Sec, the genetically encoded 21st amino acid, is biosynthesized from a reactive selenium donor on its cognate tRNA[Ser]Sec. It is thought that intracellular selenium must be delivered 'safely' and 'efficiently' by a carrier protein owing to its high reactivity and very low concentrations. Here, we identified peroxiredoxin 6 (PRDX6) as a novel selenoprotein synthesis factor. Loss of PRDX6 decreases the expression of selenoproteins and induces ferroptosis via a reduction in GPX4. Mechanistically, PRDX6 increases the efficiency of intracellular selenium utilization by transferring selenium between proteins within the selenocysteyl-tRNA[Ser]Sec synthesis machinery, leading to efficient synthesis of selenocysteyl-tRNA[Ser]Sec. These findings highlight previously unidentified selenium metabolic systems and provide new insights into ferroptosis.

Ferroptosis, a type of programmed cell death triggered by iron-induced lipid hydroperoxidation, is involved in numerous pathological conditions such as cancer, neurodegenerative diseases and ischemia–reperfusion injury; as such, it has been studied extensively[1,2]. Cells have two main defense mechanisms against ferroptosis: GPX4 and ferroptosis suppressor protein 1 (FSP1). GPX4 uses glutathione (GSH) to suppress ferroptosis by detoxifying lipid hydroperoxides into nontoxic lipid alcohols[3–5], and FSP1 is an oxidoreductase that reduces ubiquinone (CoQ) to ubiquinol (CoQH$_2$), a radical-trapping antioxidant[6,7].

GPX4 is a selenoprotein in which the highly reactive 21st amino acid Sec resides at the active site[4]. Sec is incorporated into the UGA codon when the Sec-insertion sequence (SECIS) element is present in the 3′-untranslated region of mRNAs[8–10]. Unlike other amino acids, selenocysteyl-transfer RNA for Sec (Sec-tRNA[Ser]Sec) is biosynthesized from reactive selenium donor selenide on its cognate tRNA[Ser]Sec

through intricate enzymatic reactions[8,11]. Selenide is phosphorylated by selenophosphate synthetase 2 (SEPHS2)[11–13] and subsequently incorporated into Sec-tRNA[Ser]Sec. Synthesis of selenoproteins is essential for life, as shown by studies describing embryonic lethality in mice lacking selenoprotein synthesis[14,15]. Despite its importance, however, the selenoprotein synthesis pathway remains unclear. In particular, highly reactive selenides are hypothesized to be delivered safely and efficiently by a carrier protein(s) to SEPHS2; however, they have not yet been identified[11,16].

Here, we conducted a genome-wide iron-triggered ferroptosis screen and identified PRDX6 as a novel selenoprotein synthesis factor. Loss of PRDX6 greatly decreased the expression of selenoproteins, including GPX4, and triggered ferroptosis through a reduction in GPX4 levels. Mechanistically, PRDX6 functions as a selenide carrier, transferring selenide to SEPHS2 to facilitate efficient selenium utilization.

[1]Department of Molecular and Cellular Physiology, Kyoto University School of Medicine, Kyoto, Japan. [2]Laboratory of Toxicology and Environmental Health, Graduate School of Pharmaceutical Sciences, Chiba University, Chiba, Japan. [3]Department of Environmental Medicine and Molecular Toxicology, Tohoku University Graduate School of Medicine, Sendai, Japan. [4]Present address: Department of Radiation Oncology, New York University Langone Health, New York, NY, USA. [5]These authors contributed equally: Yu-ki Tanaka, Seiryo Ogata. ✉e-mail: fujisan@mcp.med.kyoto-u.ac.jp; kiwai@mcp.med.kyoto-u.ac.jp

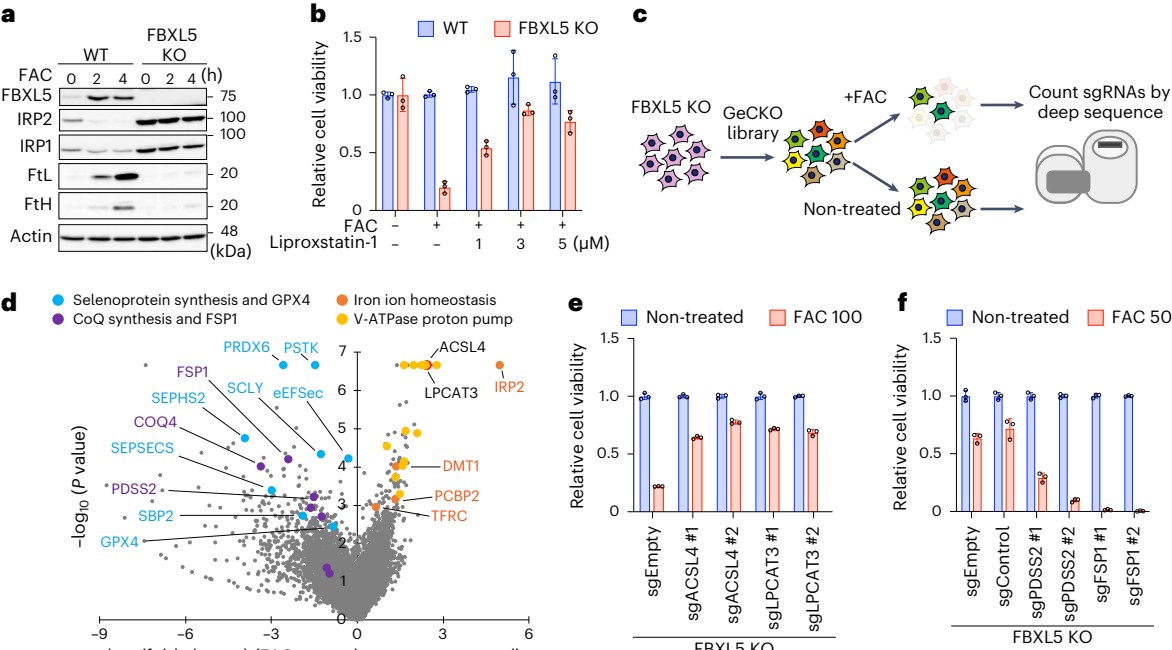

**Fig. 1 | Genome-wide CRISPR screening for iron-triggered ferroptosis. a**, WT or FBXL5 KO MEFs were treated with FAC (100 μg ml⁻¹) for the indicated periods, and cell lysates were analyzed by immunoblotting. Data are representative of three independent experiments. **b**, Viability of WT or FBXL5 KO MEFs treated for 48 h with FAC (100 μg ml⁻¹) in the presence of liproxstatin-1 (0, 1, 3 or 5 μM). **c**, Schematic showing the CRISPR–Cas9 screening strategy. **d**, Volcano plots showing the regulatory genes involved in iron-triggered ferroptosis. The $P$ value

and fold change were generated by the MAGeCK test. **e**, Viability of FBXL5 KO cells expressing sgEmpty or sgRNAs targeting ACSL4 or LPCAT3; cells were treated with FAC (100 μg ml⁻¹) for 48 h. **f**, Viability of FBXL5 KO cells expressing sgEmpty, sgControl or sgRNAs targeting PDSS2 or FSP1; cells were treated with FAC (50 μg ml⁻¹) for 30 h. Viability data (in **b**, **e**, **f**) are presented as the mean ± s.d. of three biological replicates.

Increased selenium utilization suppresses ferroptosis and contributes to cancer progression[2,17]. We found that patients harboring cancers with high expression of PRDX6 have a poor prognosis. These findings provide insight into the previously unknown selenium metabolic system in cells as well as its potential as a target for cancer therapy.

## Results

### Iron-induced ferroptosis screen

The prefix 'ferro' suggests that ferroptosis should be categorized as a form of iron toxicity; however, in most studies of ferroptosis, cell death is not induced by iron administration. Rather, it is induced by inhibiting erasers for lipid hydroperoxidation such as the GPX4 inhibitor RSL3 or the cystine transporter inhibitor erastin[3,18]. To re-analyze ferroptosis from the perspective of iron toxicity and to identify new regulators, we developed a ferroptosis induction system based on iron addition alone. The iron regulatory protein 2 (IRP2) and F-box and leucine-rich repeat protein 5 (FBXL5) system acts as a major regulator of cellular iron homeostasis by suppressing iron uptake and increasing iron storage by FBXL5-mediated degradation of IRP2 (refs. [19–21]). Therefore, to induce iron toxicity[22], we knocked out FBXL5 from mouse embryonic fibroblasts (MEFs) using the CRISPR–Cas9 system (Fig. 1a). Treatment with ferric ammonium citrate (FAC) selectively killed FBXL5 knockout (KO) cells by increasing the cellular iron concentration, as detected by a fluorescent probe specific for Fe²⁺ (Fig. 1b and Extended Data Fig. 1a,b)[23]. Ferroptosis is thought to be triggered by hydroperoxidation of polyunsaturated fatty acids (PUFAs) in the cell membrane by redox-active iron[1]. BODIPY 581/591 C11 staining to detect hydroperoxidation of PUFAs revealed increased hydroperoxidation in iron-repleted FBXL5 KO cells (Extended Data Fig. 1c). Iron-triggered death of FBXL5 KO cells appeared to occur through ferroptosis because it was prevented by the ferroptosis inhibitor liproxstatin-1 (Fig. 1b). To identify genes involved in iron-triggered ferroptosis, we performed an unbiased genome-wide

CRISPR screen in which FBXL5 KO cells were infected with a GeCKOv2 library, followed by cultivation with or without FAC (Fig. 1c). Using a cutoff of $P < 0.001$, screening identified around 150 genes encoding iron-triggered ferroptosis suppressors or activators (Fig. 1d). As expected, one of the top hit genes encoding ferroptosis activators was IRP2; indeed, stabilization of IRP2 caused by loss of FBXL5 led to accumulation of redox-active iron in cells[24] (Fig. 1a). Other prominent hits included iron homeostasis regulators such as transferrin receptor 1 (TFRC), divalent metal transporter 1 (DMT1) and the V-ATPase proton pump complex, all of which are required for iron uptake[25] (Fig. 1d). We used the V-ATPase inhibitor bafilomycin A1 to confirm that organelle acidification is essential for iron uptake and iron-triggered ferroptosis (Extended Data Fig. 1d,e). More importantly, the high-confidence hit list included the known ferroptosis regulators acyl-CoA synthetase long-chain family member 4 (ACSL4) and lysophosphatidylcholine acyltransferase 3 (LPCAT3)[26,27] (Fig. 1d), both of which are required for insertion of PUFAs into membrane phospholipids. Deletion of ACSL4 or LPCAT3 from FBXL5 KO cells restored cell viability (Fig. 1e and Extended Data Fig. 1f), further indicating that iron-triggered ferroptosis is also mediated by oxidation of PUFAs. In addition, we identified suppressor genes, the CoQ synthesis machinery and FSP1 (Fig. 1d)[6,7]. Deletion of diphosphate synthase subunit 2 (PDSS2), a component of the CoQ synthesis machinery, or FSP1 markedly increased the death of FBXL5 KO cells (Fig. 1f and Extended Data Fig. 1g); thus, lipid radical trapping by CoQH₂ is also involved in suppressing iron-triggered ferroptosis[6,7]. Interestingly, suppressor genes encoding mitochondrial electron transport chain complex I and II (CoQ oxidoreductase), but not complex III and IV, were enriched (Extended Data Fig. 1h), suggesting that CoQH₂ in mitochondria also suppresses ferroptosis. Therefore, our iron-triggered ferroptosis screen, which does not rely on conventional ferroptosis inducers such as RSL3 or erastin, is useful for identifying not only ferroptosis activators but also suppressors.

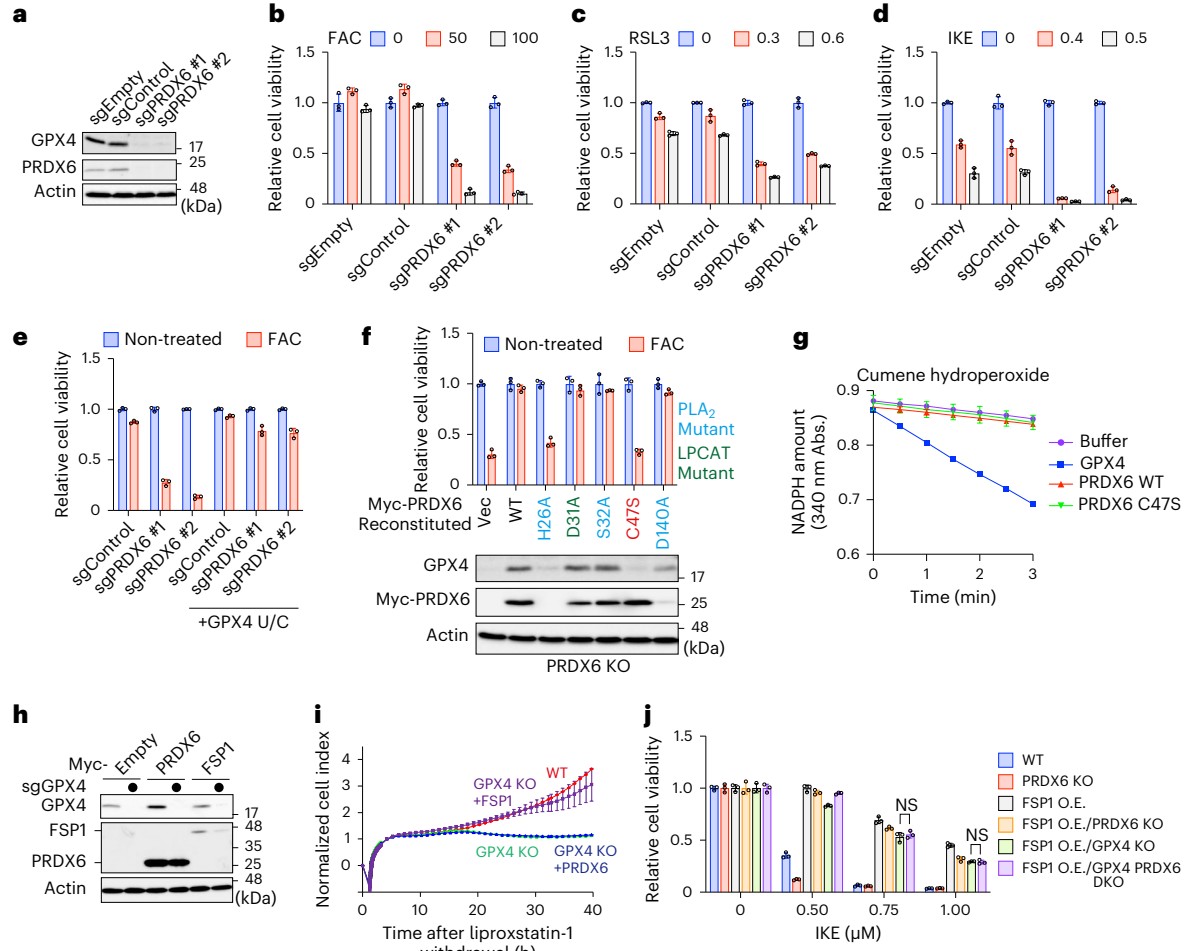

**Fig. 2 | PRDX6 suppresses iron-triggered ferroptosis indirectly by augmenting expression of GPX4. a**, Immunoblot analysis of lysates from cells expressing sgEmpty, sgControl or sgRNAs targeting PRDX6. Data are representative of three independent experiments. **b–d**, Viability of control cells or PRDX6 KO cells treated with FAC (50 or 100 µg ml⁻¹) for 48 h (**b**), RSL3 (0.3 or 0.6 µM) for 24 h (**c**) or IKE (0.4 or 0.5 µM) for 24 h (**d**). **e**, Viability of control cells or PRDX6 KO cells stably expressing the GPX4 U/C mutant; cells were incubated with FAC (50 µg ml⁻¹) for 48 h. **f**, Viability of PRDX6 KO cells stably expressing the indicated PRDX6 mutants incubated in the presence of FAC (50 µg ml⁻¹) for 48 h. Immunoblot analysis of lysates from the indicated cells is also shown. PLA₂, phospholipase A2. **g**, Measurement of GPX activity using recombinant

proteins. Cumene hydroperoxide was used as a substrate. Data are presented as the mean ± s.e.m of three independent experiments. **h**, Immunoblot analysis of lysates from control or GPX4 KO cells stably expressing the Myc-PRDX6 or FSP1. Data are representative of two independent experiments. **i**, Continuous monitoring of the viability of the indicated cell lines after withdrawal of liproxstatin-1. Data are presented as the mean ± s.e.m of three biological replicates. **j**, Viability of the indicated cells in the presence of the indicated concentrations of IKE for 24 h. NS; not significant; ns*P* (IKE 0.75 µM, *P* = 0.8885; IKE 1 µM, *P* > 0.9999); two-way ANOVA. Viability data (in **b–f,j**) are presented as the mean ± s.d. of three biological replicates.

## Identification of PRDX6 as a regulator of GPX4 expression

Gene ontology analysis of the iron-triggered ferroptosis suppressors indicated that selenoprotein synthesis factors were the most enriched group (Extended Data Fig. 2a). The high-confidence suppressor list raised the possibility that loss of selenoprotein synthesis factors sensitizes cells to iron-triggered ferroptosis (Fig. 1d). Among selenoproteins, the master ferroptosis regulator GPX4 was ranked highly as a suppressor (Extended Data Fig. 2b). Therefore, we suspected that new GPX4 regulators might be included in our suppressor list (Extended Data Fig. 2c), and we performed a secondary screen using the abundance of GPX4 as a readout (Extended Data Fig. 2d). Unexpectedly, we found that deletion of PRDX6 greatly reduced expression of GPX4 (Extended Data Fig. 2d), which we confirmed using two different guide RNAs (gRNAs) (Fig. 2a). Surprisingly, loss of PRDX6 sensitized not only FBXL5 KO cells (Extended Data Fig. 3a) but also wild-type (WT) cells (Fig. 2b) to iron-triggered ferroptosis, whereas deletion of PRDX6 had no effect on iron homeostasis because FAC-induced stabilization of FBXL5 and degradation of IRP2 was similar to that in WT cells (Extended Data

Fig. 3b)[19,20]. This finding ruled out the alternative possibility that loss of PRDX6 inhibits iron-induced ferroptosis by disrupting iron metabolism in cells. Increased iron-induced lipid hydroperoxidation in PRDX6 KO cells indicated that PRDX6 is an essential suppressor of iron-triggered ferroptosis (Extended Data Fig. 3c). In addition, we found that PRDX6 is also involved in suppressing ferroptosis mediated by imidazole ketone erastin[28] (IKE) and RSL3 (Fig. 2c,d). To address whether the decrease in GPX4 is responsible for ferroptosis sensitivity in PRDX6 KO cells, we expressed a partially active GPX4 mutant, in which Sec(U)46 is substituted by cysteine (GPX4 U/C) in PRDX6 KO cells (Extended Data Fig. 3d)[4]. Expression of the GPX4 U/C mutant rescued PRDX6 KO cells from iron-triggered ferroptosis (Fig. 2e), suggesting that PRDX6 suppresses cell death by maintaining expression of GPX4.

PRDX6 belongs to the PRDXs family of antioxidant enzymes[29–31]. Among them, PRDX6 has a unique characteristic in that it contains a single conserved cysteine (C47) residue, whereas the other family members contain two[29]. PRDX6 exhibits three enzymatic activities: glutathione peroxidase (GPX) (C47) activity, phospholipase A2

(H26, S32 and D140) activity and LPCAT (D31) activity (the residues essential for each activity are shown in parentheses)[32]. To examine which activity is required to maintain GPX4 abundance, we reconstituted PRDX6 KO cells with the WT or mutant PRDX6 (Fig. 2f). Although the H26A mutant failed to restore GPX4 levels owing to poor expression, the S32A and D140A mutants did restore them. Therefore, we concluded that phospholipase A2 activity is dispensable. We found that the C47S mutant failed to restore GPX4 expression (Fig. 2f). The C47S mutant also sensitized cells to not only iron-triggered but also canonical ferroptosis (Fig. 2f and Extended Data Fig. 3e,f). Previous reports have indicated that PRDX6 exhibits GPX activity at C47 (ref. 33); however, contrasting results have also been reported[34–36]. We found that purified PRDX6 has no GPX activity (Fig. 2g and Extended Data Fig. 3g), indicating that the conserved cysteine of PRDX6 has a different function than previously thought. Next, we asked whether PRDX6 reduces lipid hydroperoxides in cells directly. Overexpression of PRDX6 increased the expression of GPX4 in WT cells (Fig. 2h), confirming that PRDX6 is critical for the expression of GPX4. As loss of GPX4 severely sensitizes cells to ferroptosis[5], we maintained GPX4 KO cells in culture medium containing the ferroptosis inhibitor liproxstatin-1. Washout of liproxstatin-1 inhibited the growth of GPX4 KO cells (Fig. 2i). Expression of FSP1, which suppresses ferroptosis independently of GPX4, restored proliferation of GPX4 KO cells, as previously reported[6]; however, expression of PRDX6 failed to restore proliferation of GPX4 KO cells (Fig. 2i). Furthermore, expression of PRDX6 did not reduce accumulation of lipid hydroperoxides in GPX4 KO cells (Extended Data Fig. 3h), strongly indicating that PRDX6 does not suppress lipid hydroperoxidation or ferroptosis directly; rather, it suppresses them by increasing the amount of GPX4. We also generated FSP1-overexpressing (O.E.)/GPX4 and PRDX6 double knockout (DKO) cells to confirm our findings (Extended Data Fig. 3i). Loss of PRDX6 did not increase IKE-triggered ferroptosis of FSP1 O.E./GPX4 KO cells (Fig. 2j), further supporting our hypothesis that PRDX6 suppresses ferroptosis indirectly by maintaining expression of GPX4.

## Involvement of PRDX6 in selenoprotein synthesis

Next, we dissected the mechanism underlying the PRDX6-mediated increase in GPX4 expression. We found no obvious differences in the levels of GPX4 mRNA between PRDX6 KO and WT cells (Extended Data Fig. 4a), nor did proteasomal or lysosomal inhibitors restore GPX4 levels in PRDX6 KO cells (Extended Data Fig. 4b,c). Gene ontology analysis revealed enrichment of selenoprotein synthesis factors (Extended Data Fig. 2a). Moreover, we found that unbiased co-essentiality database tools[37] also indicated a strong functional connection between PRDX6 and genes involved in selenoprotein synthesis (Fig. 3a). Therefore, we reasoned that PRDX6 is involved in selenoprotein biosynthesis. Indeed, deletion of PRDX6 reduced expression of other selenoproteins, including selenoprotein N (SELN), GPX1 and SEPHS2 (Fig. 3b). The conserved Cys47 of PRDX6 is essential for expression of selenoproteins because selenoprotein expression was rescued by re-expression of PRDX6 WT, but not that of the C47S mutant, in PRDX6 KO cells (Fig. 3c). Moreover, loss of PRDX6 reduced the amount of selenoproteins in several human cancer cell lines (Extended Data Fig. 4d), indicating that the role of PRDX6 is conserved across many cancer types.

To confirm the involvement of PRDX6 in selenoprotein synthesis, we constructed cDNAs encoding model selenoproteins harboring an in-frame UGA codon at C70 or S175 of GFP (C70U and S175U), as well as the 3'UTR SECIS element of GPX4 (Fig. 3d). As expected, loss of phosphoseryl-tRNA[Ser]Sec kinase (PSTK) or SEPHS2, both of which are essential for synthesis of Sec-tRNA[Ser]Sec (ref. 38), greatly reduced expression of both GFP C70U and S175U (Fig. 3d). Loss of PRDX6 also reduced expression of both model selenoproteins (Fig. 3d), whereas loss of PRDX6 did not affect expression of GFP WT (Extended Data Fig. 4e). These results clearly indicate that PRDX6 has a crucial role in Sec insertion into the UGA codon.

## PRDX6 augments efficient utilization of selenium

Next, we explored the role of PRDX6 in selenoprotein synthesis. Both inorganic and organic forms of selenium can be used to synthesize Sec-tRNA[Ser]Sec (Extended Data Fig. 5a). Inorganic selenite is reduced by GSH and the thioredoxin system, whereas organic Sec is decomposed by selenocysteine lyase (SCLY), resulting in generation of the common active selenium donor selenide, which is important for Sec-tRNA[Ser]Sec synthesis (Extended Data Fig. 5a)[38]. Addition of excess selenite or selenocystine ($(Sec)_2$) to cultures of WT cells increased selenoprotein expression markedly, suggesting that the supply of selenium is limited under normal culture conditions (Fig. 4a). Although either selenium source failed to induce expression of selenoproteins in cells lacking PSTK or SEPHS2 (Fig. 4a), both of which are essential for Sec-tRNA[Ser]Sec synthesis[38], we found that addition of excess selenium increased the amount of selenoproteins in PRDX6 KO cells, regardless of the selenium source (Fig. 4a). Addition of selenite or $(Sec)_2$ also protected MEFs lacking PRDX6 from iron-triggered and canonical ferroptosis (Fig. 4b,c and Extended Data Fig. 5b,c). Furthermore, addition of selenium increased selenoprotein expression and suppressed iron-triggered ferroptosis in HeLa cells lacking PRDX6 (Extended Data Fig. 5d,e). These results show that addition of excess selenium can compensate for loss of PRDX6, raising the possibility that selenium is not used efficiently by PRDX6 KO cells. Indeed, when selenite or $(Sec)_2$ was added to cultures in a dose-dependent or time-dependent manner, we found that the incremental increases in selenoprotein levels were much less efficient in PRDX6 KO than in WT cells (Fig. 4d,e and Extended Data Fig. 5f,g). PRDX6 is also involved in efficient utilization of selenium after treatment with SELENOP, a more physiological source of selenium[39] (Fig. 4f). Therefore, to more rigorously evaluate the hypothesis that PRDX6 is involved in the efficiency of selenium utilization, we adjusted the initial level of selenoproteins in WT and PRDX6 KO cells and then added selenium; this is because initial expression of selenoproteins was greatly reduced in PRDX6 KO cells (Fig. 3b). We found that expression of selenoproteins in WT cells cultivated in medium with a reduced concentration of fetal bovine serum (FBS) fell markedly; FBS is the sole source of selenium in this culture system (Extended Data Fig. 5h). Next, we added selenium to culture medium containing 1% FBS and found that selenoprotein synthesis was much more efficient in WT cells than in KO cells (Extended Data Fig. 5h). Furthermore, even when PRDX6 KO cells were pretreated with selenium and then fed additional selenium, selenoprotein synthesis was lower in PRDX6 KO cells than in WT cells (Extended Data Fig. 5i). These results strongly support our hypothesis that PRDX6 is involved in efficient utilization of selenium for Sec-tRNA[Ser]Sec synthesis. Next, to evaluate the amount of Sec-tRNA[Ser]Sec in cells, we established a highly sensitive in vitro reconstructive system for selenoprotein synthesis by modifying a Sec UGA readthrough luciferase reporter assay[40]. Given that wheat germ lacks the Sec incorporation machinery[41], eukaryotic elongation factor Sec-tRNA[Ser]Sec-specific (eEFSec) and SECIS binding protein 2 (SBP2) were added to the wheat germ lysate together with a Sec-tRNA[Ser]Sec source to establish the reconstitution system (Fig. 4g). Addition of aminoacyl-tRNAs (aa-tRNAs) purified from WT or SEPHS2 KO cells (the former, but not the latter, contains Sec-tRNA[Ser]Sec) revealed that aa-tRNAs from WT cells increased selenoprotein synthesis in the presence of eEFSec and SBP2, as evaluated by measuring luciferase luminescence (Fig. 4h). Moreover, aa-tRNAs from WT cells cultivated with $(Sec)_2$ further enhanced selenoprotein synthesis. However, aa-tRNAs purified from MEFs lacking SEPHS2 did not show increased synthesis, even when KO cells were cultivated with $(Sec)_2$ (Fig. 4h), confirming the validity of the system for evaluating the amount of Sec-tRNA[Ser]Sec in aa-tRNA sources. Next, we purified aa-tRNAs from WT or PRDX6 KO cells cultured with or without $(Sec)_2$ (Fig. 4i). Purified aa-tRNAs from PRDX6 KO cells cultivated in the absence of $(Sec)_2$ failed to increase selenoprotein synthesis, and $(Sec)_2$ treatment increased it only slightly. These results clearly indicate that PRDX6 is involved in the efficient use of selenium for Sec-tRNA[Ser]Sec synthesis.

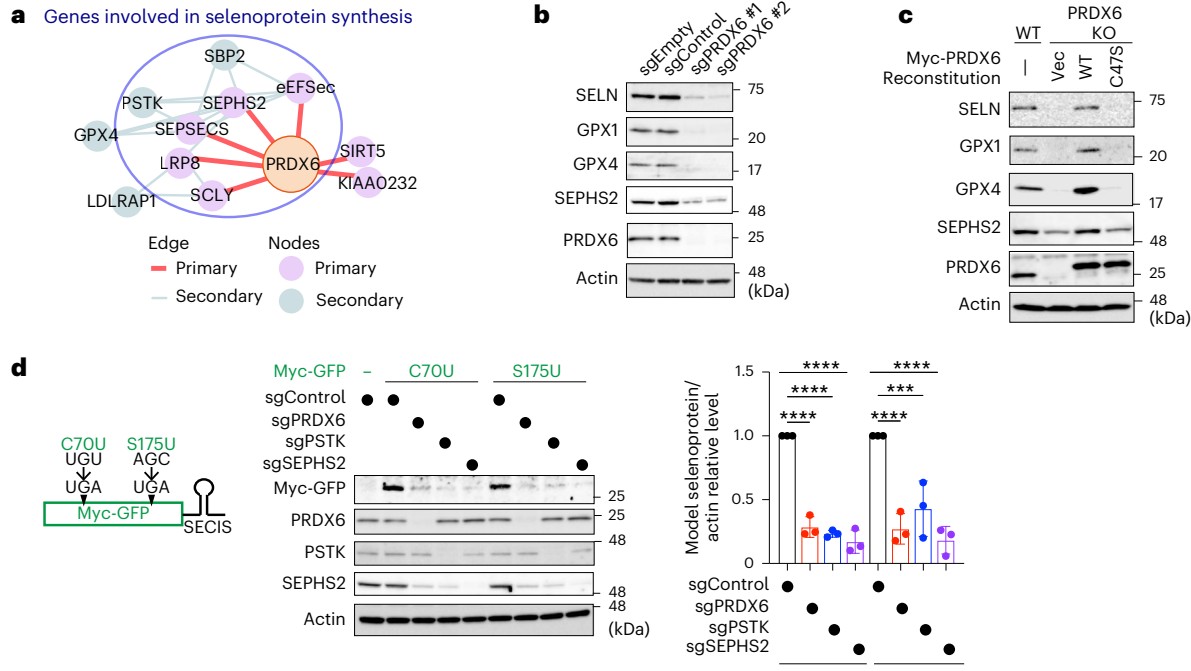

**Fig. 3 | PRDX6 is involved in selenoprotein synthesis. a**, Co-essentiality network analysis of PRDX6 using FIREWORKS. **b**, Immunoblot analysis of lysates from cells expressing sgEmpty, sgControl or sgRNAs targeting PRDX6. Data are representative of three independent experiments. **c**, Immunoblot analysis of lysates from control or PRDX6 KO cells stably expressing the PRDX6 WT or C47S mutant. Data are representative of three independent experiments.

**d**, Schematic showing the constructs of model selenoproteins (left). Immunoblot analysis of lysates from cells stably expressing Myc-GFP C70U or S175U in control cells or cells in which the indicated genes have been knocked out (middle). Quantification of the abundance of selenoproteins (right). Data are presented as the mean ± s.d. of three independent experiments. ***$P < 0.001$ ($P = 0.0001$); ****$P < 0.0001$; one-way ANOVA.

## PRDX6 augments selenium utilization as a selenide carrier

Next, we explored the mechanism(s) by which PRDX6 increases the efficiency of selenium utilization for Sec-tRNA[Ser]Sec synthesis. First, we overexpressed enzymes involved in Sec-tRNA[Ser]Sec synthesis in PRDX6 KO cells (Extended Data Fig. 5a) and found that overexpression (with the exception of SEPHS2 and PRDX6) failed to restore GPX4 expression (Fig. 5a). Thus, SEPHS2, a selenoprotein that phosphorylates selenide for Sec-tRNA[Ser]Sec synthesis[11], appears to have a critical role in GPX4 expression in PRDX6 KO cells. Previous studies have shown that a SEPHS2 mutant in which Sec is replaced by cysteine (SEPHS2 U/C) rescued the SEPHS2-deficient phenotype[42,43] (Extended Data Fig. 6a). Therefore, we overexpressed SEPHS2 U/C or WT in PRDX6 KO cells and found that SEPHS2 WT, but not the U/C mutant, restored GPX4 expression markedly (Extended Data Fig. 6b). However, we found that overexpression of SEPHS2 WT in PRDX6 KO failed to fully restore expression of other selenoproteins such as SELN and GPX1 (Extended Data Fig. 6b). We additionally found that overexpression of SEPHS2 WT also restored expression of model selenoproteins (Extended Data Fig. 6c) and protected cells from iron-triggered and canonical ferroptosis (Extended Data Fig. 6d–f). These results raise the possibility that loss of PRDX6 suppresses the expression of selenoproteins through a reduction of SEPHS2, not by facilitating utilization of selenium. To rule out this possibility, we expressed SEPHS2 WT at endogenous levels in PRDX6 KO cells because the amount of SEPHS2 in overexpressing cells was much higher than that in cells expressing endogenous SEPHS2 (Fig. 5a). We found that the level of endogenous SEPHS2 WT failed to reverse loss of selenoprotein expression in PRDX6 KO cells (Fig. 5b) and did not suppress iron-triggered and canonical ferroptosis (Extended Data Fig. 6d–f). In addition, when PRDX6 KO cells were pretreated with selenite or (Sec)₂ to adjust SEPHS2 expression in KO cells to levels comparable to those in WT cells before the time course experiment, we observed that WT cells utilized selenium more efficiently than PRDX6 KO cells

(Extended Data Fig. 5i). These results show clearly that PRDX6 is necessary for effective utilization of selenium to generate Sec-tRNA[Ser]Sec in cells expressing endogenous levels of SEPHS2. Importantly, we also found that Sec-tRNA[Ser]Sec levels were still lower in PRDX6 KO cells overexpressing SEPHS2 than in WT cells (Fig. 5c), suggesting that overexpression of SEPHS2 cannot compensate fully for the reduced efficiency of selenium utilization owing to loss of PRDX6. Collectively, the data suggest that excess selenium as well as overexpression of SEPHS2 WT, but not the U/C mutant, compensates for the loss of PRDX6, albeit not very effectively. Thus, these results imply that SEPHS2 itself can acquire selenium through the Sec residue of SEPHS2; however, PRDX6 greatly facilitates the utilization of selenium by SEPHS2.

Owing to its high reactivity, it was thought that selenide must associate with an unidentified carrier protein through a cysteine residue for efficient transfer to SEPHS2 (refs. 11,16). Addition of excess (Sec)₂ was toxic to cells; however, we found that PRDX6, in which C47 is critical, suppresses (Sec)₂ toxicity (Extended Data Fig. 6g). Considering that C47 of PRDX6 also has a crucial role in the efficient synthesis of selenoproteins (Figs. 2f and 3c), it is highly likely that PRDX6 is a potential selenide carrier. A previous report showed that rhodanese forms perselenide (-S-SeH) at cysteine residues in the presence of selenite and GSH[44]. Therefore, we incubated recombinant PRDX6 WT or C47S proteins with selenite and GSH, followed by the detection of PRDX6-bound selenium by inductively coupled plasma mass spectrometry (ICP–MS) (Extended Data Fig. 6h). We found that PRDX6 WT binds to selenium more effectively than the C47S mutant. Next, we performed mass spectrometry analyses to further characterize the PRDX6–selenium intermediate. The results showed that C47 of PRDX6 forms a perselenide bond when reacting with GSH and selenite, as well as with SCLY and Sec (Extended Data Fig. 7). Moreover, we found that about 10% of PRDX6 C47 formed a perselenide bond in a 5 min reaction with selenite and GSH, or with SCLY and Sec (Fig. 5d,e). Selenide carrier proteins

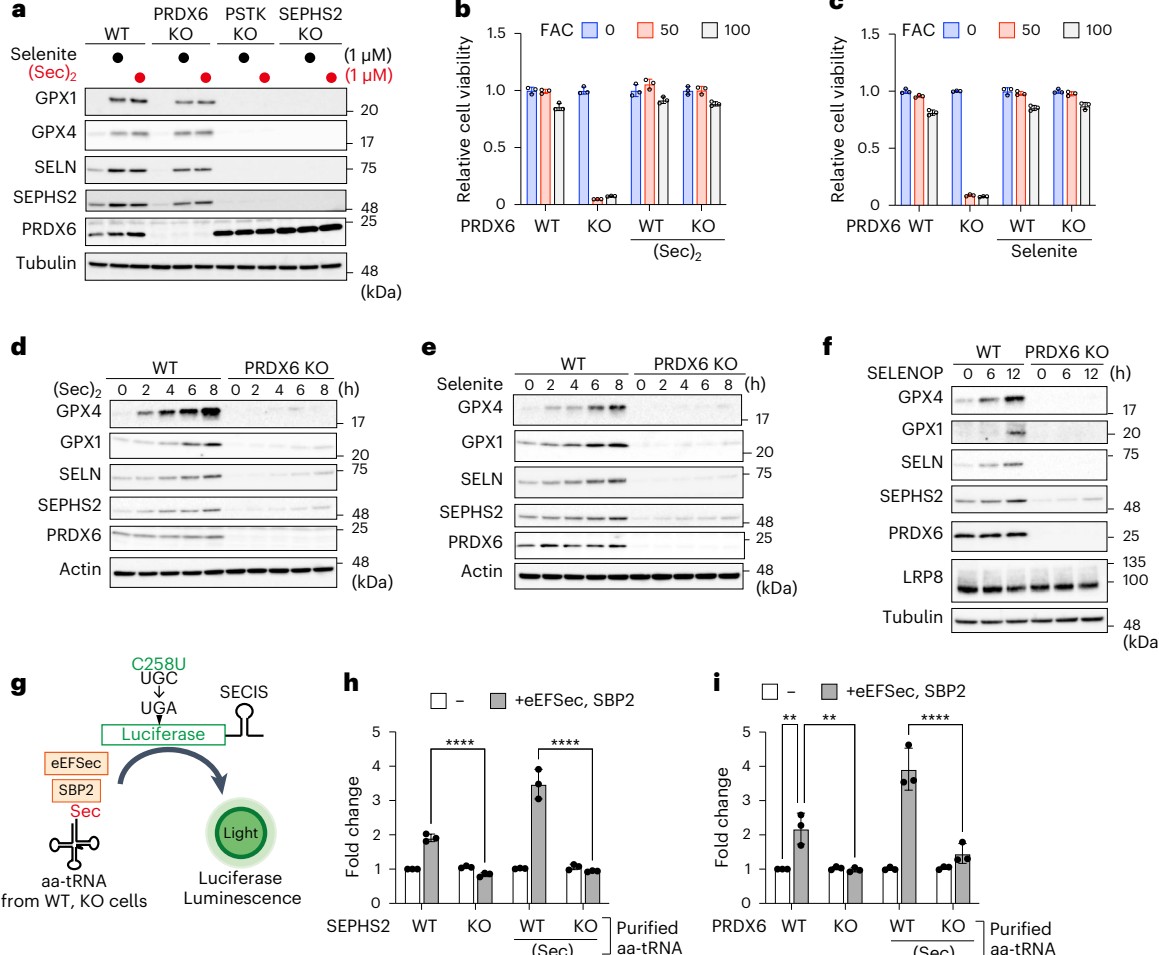

**Fig. 4 | PRDX6 augments efficient selenium utilization. a,** Immunoblot analysis of lysates from control or the indicated KO cells cultured for 32 h in the presence of sodium selenite (1 μM) or (Sec)₂ (1 μM). Data are representative of two independent experiments. **b,c,** WT or PRDX6 KO cells were pretreated for 2 days with (Sec)₂ (100 nM) or sodium selenite (100 nM). Then, cells were treated for 48 h with FAC (50, 100 μg ml⁻¹) in the presence of (Sec)₂ (100 nM) (**b**) or sodium selenite (100 nM) (**c**), and viability was measured. Viability data are presented as the mean ± s.d. of three biological replicates. **d–f,** Time course of selenoprotein expression by WT or PRDX6 KO MEFs in the presence of 50 nM (Sec)₂ (**d**), 100 nM sodium selenite (**e**) or 0.5 μg ml⁻¹ SELENOP (**f**). Data are representative of two

(in **f**) or three (in **d** and **e**) independent experiments. **g,** Schematic showing the in vitro reconstruction system used to evaluate the amount of Sec-tRNA^[Ser]Sec. **h,i,** aa-tRNAs, which were purified from the indicated cells treated (or not) for 105 min with 50 nM (Sec)₂, followed by treatment for 15 min with cycloheximide, were added to wheat germ extract in the presence or absence of eEFSec and SBP2. Data of aa-tRNAs from WT or SEPHS2 KO cells (**h**), and WT or PRDX6 KO cells (**i**) are shown. Data are presented as the mean ± s.d. of three independent experiments. **P < 0.01 (WT ± eEFSec and SBP2, P = 0.0037; PRDX6 WT versus KO, P = 0.0031); ****P < 0.0001; two-way ANOVA.

should receive selenide from SCLY and then transfer it to SEPHS2 for phosphorylation (Extended Data Fig. 5a). Proximity ligation assays (PLAs) revealed that endogenous PRDX6 bound to both SCLY and SEPHS2 effectively (Fig. 5f and Extended Data Fig. 8a). To confirm binding of PRDX6 to SCLY and SEPHS2, we expressed HA-TurboID-PRDX6 or HA-TurboID-PRDX1 (a paralog of PRDX6) in WT or PRDX6 KO MEFs. As shown in Fig. 5g and Extended Data Fig. 8b,c, PRDX6 bound specifically to SEPHS2 and SCLY.

Selenide is delivered to and phosphorylated by SEPHS2 to generate selenophosphate for Sec-tRNA^[Ser]Sec synthesis[11]. Given that selenophosphate is an unstable product[45], we evaluated the production of AMP, a by-product of selenophosphate synthesis, to evaluate the acceleration of selenophosphate synthesis by PRDX6. We used a recombinant SEPHS2 U/C protein that retains enzyme activity for selenophosphate synthesis[42,46] instead of SEPHS2 WT because expression of SEPHS2 WT is extremely low (Fig. 5a) and it is difficult to purify. In the selenophosphate synthesis reaction in which SCLY-mediated decomposition of Sec is used as a selenium source, addition of PRDX6 WT efficiently increased the synthesis of

selenophosphate, whereas C47S did not (Fig. 5h). When PRDX6 WT or C47S was preincubated with selenide to generate the PRDX6–selenide complex as a selenium source, PRDX6 WT accelerated SEPHS2 U/C-mediated production of AMP (Fig. 5i). Collectively, these data suggest that PRDX6 facilitates Sec-tRNA^[Ser]Sec synthesis by functioning as a selenide carrier protein (Fig. 5j).

## PRDX6 and cancer

Most selenoproteins have a role in maintaining redox balance by reducing oxidative insults, and high expression of selenoproteins is associated with malignancy grade[8,43]. Consistent with the relationship between selenoproteins and cancer, database analyses revealed that expression of PRDX6 is higher in cancerous tissues than in normal tissues[47] (Extended Data Fig. 9a) and that high expression of PRDX6 correlates with a poor prognosis for various cancers[48] (Extended Data Fig. 9b). A previous report showed that intractable pancreatic cancer cells are sensitive to ferroptosis[49]. Database analysis also revealed that expression of PRDX6 is high in pancreatic cancer and that high expression correlates with a poor prognosis (Extended Data Fig. 9).

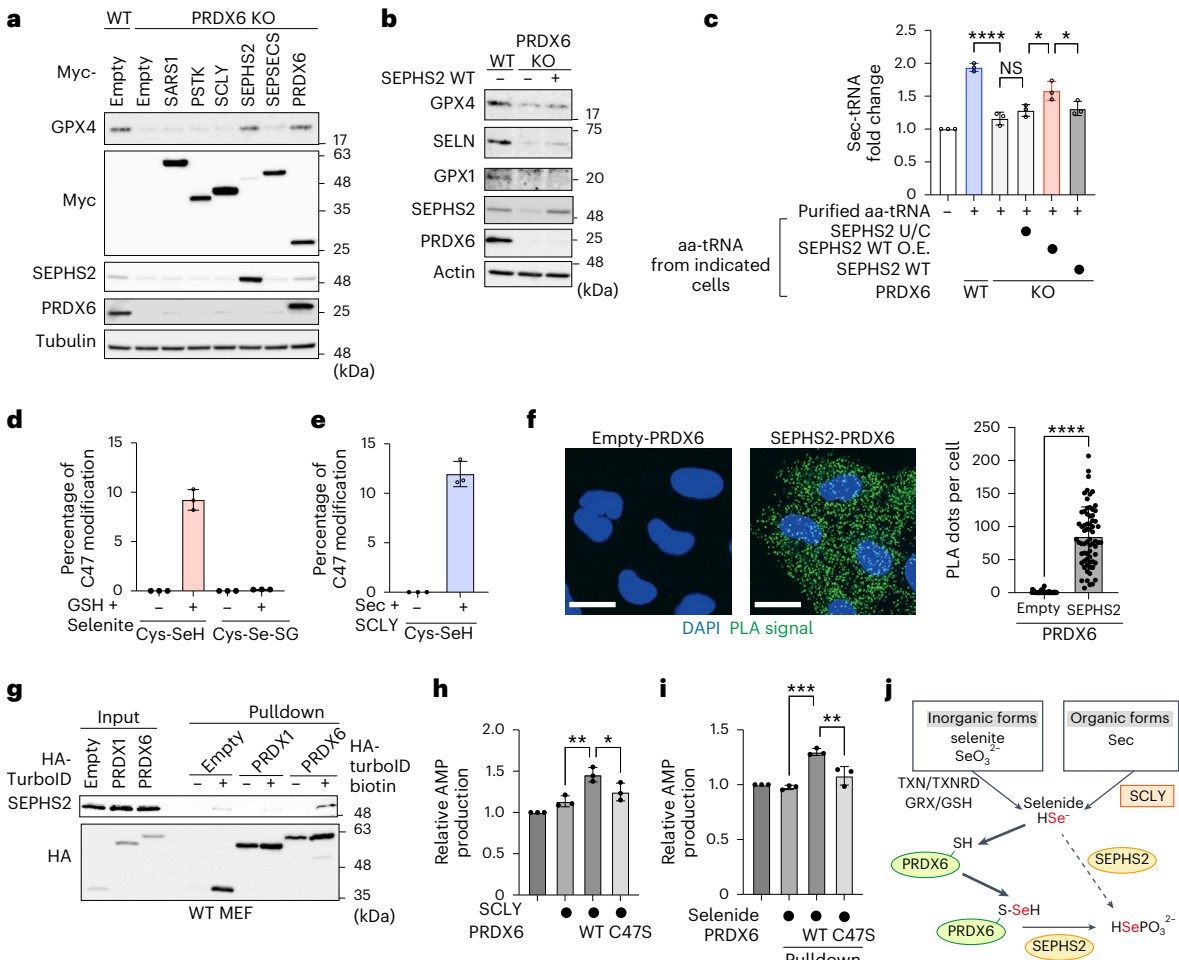

**Fig. 5 | PRDX6 is a selenide carrier protein that enables efficient selenium utilization. a**, Immunoblot analysis of lysates from control or PRDX6 KO cells stably expressing enzymes involved in Sec-tRNA[Ser]Sec synthesis. Data are representative of three independent experiments. **b**, Immunoblot analysis of lysates from control or PRDX6 KO cells stably expressing SEPHS2 WT at endogenous levels. Data are representative of three independent experiments. **c**, aa-tRNAs, purified from the indicated cells treated with cycloheximide for 15 min, were added to wheat germ extract in the presence of eEFSec and SBP2. Data are expressed as the mean ± s.d. of three independent experiments. ****$P < 0.0001$; *$P < 0.05$ (SEPHS2 U/C versus SEPHS2 WT O.E., $P = 0.0213$; SEPHS2 WT O.E. versus SEPHS2 WT, $P = 0.0422$); NS ($P = 0.6212$); one-way ANOVA. **d,e**, Percentage of C47 modification as calculated by mass spectrometry analysis. PRDX6 was incubated with GSH and sodium selenite (**d**), or with Sec and SCLY (**e**) for 5 min before mass spectrometry analyses. Data are presented as the mean ± s.d of three biological replicates. **f**, The physical association

between PRDX6 and SEPHS2 was detected in a PLA. Scale bars, 20 μm. Data are representative of three independent experiments. Quantification of the PLA dots is also shown. Data are presented as the mean ± s.d. ($n = 42$ cells (empty-PRDX6), $n = 67$ cells (SEPHS2-PRDX6) examined), ****$P < 0.0001$; unpaired two-sided $t$-test. **g**, MEFs expressing HA-TurboID-empty, -PRDX1 or -PRDX6 were cultured for 30 min with DMSO or 50 μM biotin. Cell lysates and pulldown samples were analyzed by immunoblotting. Data are representative of two independent experiments. **h,i**, PRDX6-mediated enhancement of selenophosphate-mediated synthesis of SEPHS2 was evaluated by measuring the AMP product. Data are presented as the mean ± s.d of three independent experiments in which selenocysteine and SCLY were used as the selenium source; *$P < 0.05$ ($P = 0.0423$); **$P < 0.01$ ($P = 0.004$); one-way ANOVA (**h**) or selenium bound to PRDX6 was used as the selenium source; **$P < 0.01$ ($P = 0.0018$); ***$P < 0.001$ ($P = 0.0001$); one-way ANOVA (**i**). **j**, Schematic showing the role of PRDX6 as a selenide carrier.

We found that loss of PRDX6 from two pancreatic cancer cell lines suppressed growth, a phenomenon reversed by the addition of selenium or liproxstatin-1 (Extended Data Fig. 10a–d). To consolidate our findings, we obtained several cancer cells and evaluated their sensitivity to iron-triggered ferroptosis (Extended Data Fig. 10e,f). The results showed that MYC-N-amplified neuroblastoma cells (SK-N-DZ and NB-1) were sensitive to iron-induced ferroptosis (Extended Data Fig. 10e,f). We also found that loss of PRDX6 from SK-N-DZ cells suppressed their growth markedly (Extended Data Fig. 10g,h). A previous study reported that MYC-N increases intracellular iron by upregulating the expression of TFRC, a protein required for iron uptake[50]. In addition, cancer cells are generally addicted to iron, which drives proliferation. Thus, inhibiting PRDX6 may be an effective strategy for sensitizing cancer cells specifically to iron-triggered ferroptosis.

## Discussion

Here, we developed a ferroptosis induction system that can kill cells upon addition of iron to the culture medium (Fig. 1c,d). We used this system to conduct a genome-wide CRISPR screen and identified PRDX6 as a selenoprotein synthesis factor. Although PRDX6 has long been considered an antioxidant enzyme[29,31], we show here that it exerts antioxidant effects indirectly by facilitating the expression of selenoproteins. Although organic and inorganic selenium are metabolized to selenide, which is then used for selenoprotein synthesis[38], the metabolic pathway has not yet been shown conclusively. Our results indicate that PRDX6 is an unidentified selenide carrier protein that facilitates the efficient utilization of selenide by SEPHS2 (Figs. 4 and 5).

Among the PRDX family, which comprises six proteins (PRDX1–PRDX6), the two-cysteine proteins PRDX1–PRDX5 harbor a disulfide

bond between the two conserved cysteine residues, which is reduced by the thioredoxin system. The one-cysteine protein PRDX6 is unique in that it is thought to be reduced by GSH and to have GPX activity[29,30]. However, we showed conclusively in this study that PRDX6 does not possess GPX activity (Fig. 2g and Extended Data Fig. 3g). Instead, we found that a C47S mutant of PRDX6 profoundly reduces the amount of selenoproteins, including GPX1 and GPX4 (Fig. 3c), both of which are well-known proteins that exhibit GPX activity. Given that the GPX function of PRDX6 has been evaluated mainly using PRDX6 knockdown or KO cells[51–53], the reduced GPX activity in PRDX6-deficient cells might be caused by a reduction in GPX1 and GPX4. Consistent with our results (Fig. 2g and Extended Data Fig. 3g), other reports used purified proteins to show that PRDX6 has no GPX activity[34–36]. Therefore, we conclude that PRDX6 is not an antioxidant enzyme like PRDX1–PRDX5 but it does function as a selenide carrier protein. This was confirmed by our finding that PRDX1, an antioxidant enzyme that is considered to be a paralog of PRDX6, did not restore GPX4 expression in PRDX6 KO MEFs (Extended Data Fig. 8b).

Both inorganic and organic selenium function as sources for SEPHS2 to generate selenoproteins[38]. Although the mechanism underlying delivery of selenium to SPEHS2 remains unknown, we clearly showed that PRDX6 facilitates the synthesis of selenoproteins (Fig. 4d–f and Extended Data Fig. 5f–i) and forms a perselenide bond at the C47 of both selenium sources (Fig. 5d,e and Extended Data Fig. 7). We also found that PRDX6 binds to both SEPHS2 and SCLY; the latter is an enzyme that releases selenide from Sec. Thus, PRDX6 appears to have a crucial role in delivering selenide from organic selenium by accepting selenide from SCLY and delivering it to SEPHS2. Although the mechanisms underlying the metabolism of inorganic selenium and its delivery to SEPHS2 remain unknown, our results strongly suggest that PRDX6 also has a role in the delivery of selenide to SEPHS2, even from inorganic selenium. Therefore, the identification of PRDX6 as a selenium carrier protein opens new avenues of research into intracellular selenium dynamics.

It is noteworthy that overexpression of SEPHS2 WT substantially restored the expression of selenoproteins in PRDX6 KO cells (Fig. 5a and Extended Data Fig. 6b,c). These observations imply that SEPHS2 can bind directly to selenide to produce selenophosphate in the absence of PRDX6, albeit much less efficiently; indeed, endogenous levels of SEPHS2 failed to restore selenoprotein expression. Given that the SEPHS2 U/C mutant could not restore expression of selenoproteins in PRDX6 KO cells (Extended Data Fig. 6b,c), the Sec residue in SEPHS2 seems to facilitate utilization of selenide, although the precise molecular mechanism remains unsolved. The Sec residue and C47 of PRDX6 have a low p$K_a$ and high reactivity[29,54]; therefore, we suspect that highly reactive residues may be important for the reaction with selenide in cells.

We also observed that addition of excess selenium (both organic and inorganic forms) restored the amount of selenoproteins in PRDX6 KO cells (Fig. 4a–c). Therefore, the ability of SEPHS2, which reacts directly with selenide to induce the synthesis of some selenoproteins, may underlie the finding of trace amounts of selenoproteins in PRDX6 KO cells. These observations appear to be consistent with differences in the phenotypes of PRDX6 KO mice and mice lacking selenoprotein synthesis[14,15,55,56]. Two lines of PRDX6 KO mice have been reported[55,56]. Both are viable and fertile, and neither shows an overt phenotype, although they are hyper-sensitive to various oxidative stresses. By contrast, mice lacking selenoprotein synthesis are embryonic lethal[14,15]. The observation that loss of PRDX6 does not completely attenuate selenoprotein synthesis may be advantageous with respect to the development of PRDX6 inhibitors as anti-cancer drugs. Effective inducers of ferroptosis are believed to be a promising strategy for anti-cancer therapy, and GPX4 is regarded as a suitable target[3]. However, inhibition of GPX4 may have severe side effects because GPX4 KO mice are embryonic lethal[57]. Taken together, the present results suggest that PRDX6 may be an alternative target for anti-cancer drugs, with fewer side effects.

## Online content

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

## Methods

### RT–PCR and plasmids

The open reading frames of mouse (m) PRDX6, PRDX1, SEPHS2, GPX4 (cytosolic form), SARS1, PSTK, SEPSECS, SCLY and FSP1, and human (h) PRDX6, SEPHS2, SCLY, eEFSec, SBP2 and SEPHS2-3'UTR(SECIS) were amplified by RT–PCR. Single-guide RNA (sgRNA) resistant constructs of mPRDX6 and mSEPHS2, and mutants of mPRDX6 (H26A, D31A, S32A, C47S and D140A), hPRDX6 C47S, mGPX4 U46C, mSEPHS2 U/C and hSEPHS2 U/C were generated by two-step PCR. GFP-3'SECIS (WT, C70U and S175U) was generated from the amplified open reading frames of GFP and the 3'-SECIS of GPX4. DNAs were ligated into the appropriate epitope-tag sequences and then cloned into pMXs-IP, pMXs-IRES-Bsr, pMXs-neo and pT7-7 vectors. Luciferase C258U-3'SECIS was generated from the luciferase open reading frame and the 3'-SECIS of GPX4 and then ligated into the pEU vector. CRISPR sgRNA sequences were selected from CHOPCHOP or Benchling. The oligonucleotide sequences preceding the protospacer motif were as follows:

mFBXL5, GAGGAAGATGGGTTATACCTG; mACSL4 #1, GGTTC-TACGGGCCGCCCCAA; mACSL4 #2, GACCGATCACAATCTCACCTC; mLPCAT3 #1, GCCGGTGACTACGATATCAAGTGG; mLPCAT3 #2, GTAATAACGGCAGTGACGGTGCGG; mPDSS2 #1, GATGCCGGCT-GTCGTGCACGA; mPDSS2 #2, GCGGCATAACCTACAACTGCG; mFSP1 #1, GCCAGCGCTCACAATTCATCG; mFSP1 #2, GCGCTCACAATTCATCGTGG; mPRDX6 #1, GATAGAACTATACCTTGCTCC; mPRDX6 #2, GCTCAC-CACACGGGCCGTCAC; mPSTK, GAAAGTCGACTTTCCGGCCGC; mSEPHS2, GAACCCGTGGATTATCATCGG; mMETAP1, GTGTACCGATAGC-CTGCCCA; mSNX27, GTTGACAATGCGCACGACCCG; mUBE2N, GTGGG-GACCACTTATCTATGA; mCINP, GGCCCCCTCTATTTCACACG; mCENPE, GCGCTATTTATCAGAGCGATG; mSTX5A, GTATGCGACTCGATGATCCCG; mDDX10, GAGCGATTAAAGCTCCGCACC; mKLF5, GACCTGAGGACT-CATACGGGT; mSCLY, GTAATGAGACCGGCGTCATCA; mGPX4, GAC-GATGCACACGAAACCCC; control gRNA, GCGAGGTATTCGGCTCCGCG; hPRDX6, GATGCGGCCGACGGTGGTAT. Guide sequences were inserted into pX459 (ref. 58) or lentiCRISPRv2 (ref. 59) (puro, bsr or hygro).

### Antibodies and reagents

The following antibodies were used: anti-PRDX6 (Proteintech, 13585-1-AP; western blotting (WB), 1:2,000; PLA, 1:100), anti-SEPHS2 (Proteintech, 14109-1-AP; WB, 1:2,000), anti-GPX4 (Proteintech, 67763-1-AP; WB, 1:2,000), anti-FSP1 (Proteintech, 20886; WB, 1:2,000), anti-SCLY (Proteintech, 67606-1-Ig; WB, 1:2,000), anti-GPX4 (Santa Cruz, sc-166570; WB, 1:2,000), anti-SELN (Santa Cruz, sc-365824; WB, 1:2,000), anti-GPX1/2 (Santa Cruz, sc-133160; WB, 1:2,000), anti-PSTK (Santa Cruz, sc-373991; WB, 1:2,000), anti-FBXL5 (Santa Cruz, sc-390102; WB, 1:2,000), anti-FTH1 (Santa Cruz, sc-376594; WB, 1:300), anti-ACSL4 (Santa Cruz, sc-365230; WB, 1:2,000), anti-PDSS2 (Santa Cruz, sc-515137; WB, 1:2,000), anti-ferritin (Sigma-Aldrich, F6136; WB, 1:2,000), anti-IRP2 (in-house; WB, 1:1,000), anti-LRP8 (Abcam, ab108208; WB, 1:2,000), anti-β-actin (Sigma-Aldrich, A5316; WB, 1:15,000), anti-tubulin (CEDARLANE, CLT9002; WB, 1:5,000), anti-Myc (Merck, 05-724; WB, 1:2,000; PLA, 1:200), anti-HA (MBL, M180-3; WB, 1:2,000), anti-ubiquitin K48-specific (Merck, ZRB2150; WB, 1:2,000), anti-p62 (Wako, 018-22141; WB, 1:2,000), HRP-linked anti-mouse IgG (Cell Signaling, 7076; WB, 1:10,000) and HRP-linked anti-rabbit IgG (GE Healthcare, NA934; WB, 1:10,000).

The following reagents were also used: ferric ammonium citrate (Sigma-Aldrich, F5879), liproxstatin-1 (Selleck, S7699), bafilomycin A1 (Selleck, S1413), IKE (Selleck, S8877), RSL3 (Selleck, S8155), E64d (Peptide Institute, 4321-v), pepstatin A (Peptide Institute, 4397-v), sodium selenite (Sigma-Aldrich, 214485), selenocystine (Tokyo Chemical Industry, E1368), cycloheximide (ALBIOCHEM, 239764), hydrogen peroxide (Santoku Chemical Industry, 18412), tert-butyl hydroperoxide (Sigma-Aldrich, 458139), Tris(2-carboxyethyl)phosphine (TCEP) (Nacalai, 07277), pyridoxal-5-phosphate (PLP) (Sigma-Aldrich, P9255) and biotin (WAKO, 021-08712).

### Cell lines and cell culture

MEFs were generated in-house. HepG2 was gifted by K. Nakajima (Osaka City University), originally purchased from ATCC (HepG2: HB-8065). HEK293T was gifted by E. Nakamura (Kyoto University), originally purchased from RIKRN RBC (293T: RCB2202). PLATE was gifted by T. Kitamura (Tokyo University). A549, H226, H460 and H1975 were gifted by A. Sato (Kyoto University), originally purchased from ATCC (A549: CCL-185, H226: CRL-5826, H460: HTB-177, H1975: CRL-5908). SK-N-DZ and HeLa cells were purchased from ATCC (SK-N-DZ: CRL-2149, HeLa: CCL-2). PANC-1 and MIA Paca-2 cells were purchased from RIKEN RBC (PANC-1: RCB2095, MIA Paca-2: RCB2094). NB-1 cells were purchased from JCRB (NB-1: JCRB0621). All cells were cultured in a humidified incubator at 37 °C and 7.5% $CO_2$. MEFs, HEK293T, PLATE, A549, HeLa, PANC-1, MIA Paca-2 and HepG2 cells were grown in DMEM supplemented with 10% FBS, 100 IU $ml^{-1}$ penicillin and 100 μg $ml^{-1}$ streptomycin. SK-N-DZ cells were grown in DMEM supplemented with 10% FBS, 1× non-essential amino acids (Gibco, 11140050), 100 IU $ml^{-1}$ penicillin and 100 μg $ml^{-1}$ streptomycin. GPX4, SEPHS2 and PSTK KO MEF cells were maintained in medium containing 2 μM of liproxstatin-1. H226, H460 and H1975 cells were grown in Roswell Park Memorial Institute (RPMI) 1640 medium supplemented with 10% FBS, 100 IU $ml^{-1}$ penicillin and 100 μg $ml^{-1}$ streptomycin. NB-1 cells were grown in 45% RPMI and 45% MEM medium supplemented with 10% FBS, 100 IU $ml^{-1}$ penicillin and 100 μg $ml^{-1}$ streptomycin.

### Generation of FBXL5 KO MEFs

To generate FBXL5 KO MEFs, a NEPA21 electroporator (NEPAGENE) was used to electroporate MEFs with the pX459 plasmid containing an sgRNA sequence specific for mFBXL5. Cells were then treated with puromycin for 2 days. Following selection, the cells were seeded at a low density and isolated colonies were picked. To identify FBXL5 KO cells, expression of FBXL5 was analyzed by immunoblotting.

### Lentivirus production and generation of CRISPR–Cas9-mediated KO cell lines

Lentivirus was produced by co-transfecting HEK293T cells with the LentiCRISPRv2 (puro, bsr or hygro)-containing guide sequences, the psPAX2 packaging plasmid and the VSV-G envelope plasmid using PEI MAX (Polysciences) transfection reagent. On the following day, the medium was replaced with fresh medium. After 2 days of culture, lentivirus-containing supernatant was collected and used to infect target cells overnight in the presence of 10 μg $ml^{-1}$ polybrene (Merck). Infected cells were selected with puromycin, blasticidin or hygromycin.

### Retroviral expression

pMXs-IP, pMXs-neo or pMXs-IRES-Bsr containing appropriate inserts were transfected into PLATE packaging cells or GP2-293 cells along with the pVSV-G plasmid. The resultant viruses were used to infect target cells in the presence of 10 μg $ml^{-1}$ polybrene. Stably transduced cells were selected using puromycin, G-418 or blasticidin.

### Cell lysis and western blotting

Cells were washed with PBS and lysed with lysis buffer (50 mM Tris-HCl (pH 7.5), 150 mM NaCl, 1% Triton X-100, 1 mM PMSF, and protease inhibitor cocktail (Sigma-Aldrich)). The lysates were clarified by centrifugation at 20,400$g$ for 20 min at 4 °C. Protein concentrations were determined using the Bradford assay (Nacalai Tesque), and equal amounts of protein were mixed with SDS sample buffer. Samples were heated for 5 min at 95 °C, separated by SDS–PAGE and transferred onto PVDF membranes (Millipore). After blocking in Tris-buffered saline (TBS) containing 0.1% Tween-20 and 5% (w/v) nonfat dry milk, the membrane was incubated with the appropriate primary antibodies followed by appropriate secondary antibodies. The membranes were visualized using enhanced chemiluminescence and analyzed on a LAS4000mini or LAS3000 instrument (GE Healthcare).

## Cell viability assay

Cells were plated in 96-well plates (in triplicate and at a density of 2,500–10,000 cells per well) in the presence or absence of FAC, RSL3 or IKE. After culture for 24–48 h, 10 µl of Cell Counting Kit-8 (DOJINDO) was added for 1–2 h. Absorbance at 450 nm was measured using a SpectraMax M5 microplate reader (Molecular Devices). Cell viability, monitored continuously using the iCELLigence or xCELLigence system (ACEA Bioscience), was expressed as an impedance-based cell index. GPX4 KO cells (4,500 cells per well) were plated onto an E-Plate L8 PET (ACEA Bioscience), and pancreatic ductal adenocarcinoma cells (3,000 cells per well) and SK-N-DZ cells (6,000 cells per well) were plated on an E-Plate 16 PET (ACEA Bioscience). SK-N-DZ cells were cultured in DMEM supplemented with 10% FBS, 1× non-essential amino acids and 0.25 mM L-glutamine (Fujifilm). The cell index was monitored continuously. Recording data were analyzed by RTCA software lite v.2.2.1.

## Cell staining using BODIPY 581/591 C11, and iron staining

For BODIPY 581/591 C11 (Invitrogen) staining, cells were plated in a six-well plate and treated with FAC. After 1.5–24 h, the medium was removed and the cells were labeled with DMEM containing 10 µM BODIPY 581/591 C11 at 37 °C for 30 min. Cells were then washed three times with PBS and detached from the plate using trypsin. Initial cell population gating (FSC-Area versus FSC-height) was used to ensure doublet exclusion, and green fluorescence was measured by using FACS Canto II (BD Biosciences) and FACS Diva software v.6.1.2 (Becton Dickinson). Data were analyzed using FlowJo software (v.9.9.6). For iron staining, cells were plated in a 35-mm glass bottom dish and treated with 25 µg ml⁻¹ FAC for 4 h. Cells were then washed three times with HBSS and labeled at 37 °C for 30 min with 1 µM FerroOrange (DOJINDO) followed by visualization under a Fv1000 confocal microscope (Olympus).

## Genome-wide CRISPR screening

FBXL5 KO MEFs were infected with the mouse GeCKO v.2 library[59] at a multiplicity of infection of 0.3 and selected with puromycin for 1 week post infection. The cells were then treated with 100 µg ml⁻¹ FAC for 48 h and cultured for a further 24 h in fresh medium. The cells were lysed in NTE buffer (15 mM Tris-HCl (pH 7.5), 150 mM NaCl and 1 mM EDTA), and genomic DNA from non-treated and FAC-treated cells was prepared by phenol-chloroform extraction and isopropanol precipitation. sgRNA sequences were amplified from genomic DNA by PCR using Herculase II Fusion DNA polymerase. The resultant amplicons were gel-extracted and subjected to DNA sequencing on a Novaseq 6000 (Illumina) sequencer. Sequence data were analyzed using the MAGeCK pipeline.

## PLA

The PLA was conducted using the Proximity Ligation Kit (Sigma-Aldrich). A549 cells, either empty or expressing Myc-SEPHS2 or Myc-SCLY, were cultured on micro cover glass slips in a 6-well plate. Cells were washed with PBS and fixed for 20 min with 4% formaldehyde in PBS at room temperature (25–27 °C). Cells were then washed twice with PBS and permeabilized with 0.1% Triton X-100 in PBS at room temperature for 10 min. Cells were washed twice with PBS and then incubated at 37 °C for 1 h with Duolink blocking solution. Cells were then incubated at room temperature for 1 h with primary antibodies targeting PRDX6 (rabbit polyclonal; 1:100) and Myc (mouse monoclonal; 1:200). Cells were washed twice for 5 min with Duolink wash buffer A at room temperature and then incubated with a secondary antibody (anti-mouse minus and anti-rabbit plus) at 37 °C for 1 h. Cells were washed twice at room temperature for 5 min with Duolink wash buffer A and then incubated with ligase solution at 37 °C for 30 min. Next, the cells were washed twice with Duolink wash buffer A and incubated with polymerase in amplification buffer at 37 °C for 100 min. Cells were washed twice at room temperature for 10 min with Duolink wash buffer B, followed by 0.01% Duolink wash buffer B for 1 min. Finally, the cover glasses were mounted with Duolink PLA mounting medium with DAPI. Protein–protein interactions were visualized under an Fv1000 confocal microscope (Olympus). PLA foci were counted by ImageJ (v.2.3.0).

## TurboID pulldown

Cells were treated (or not) for 30 min with 50 µM biotin, washed with PBS and lysed with lysis buffer (50 mM Tris-HCl (pH 8.0), 150 mM NaCl, 1% Triton X-100 and 1 mM PMSF). The lysates were clarified by centrifugation at 20,400g for 20 min at 4 °C. Protein concentrations were determined using the Bradford assay, and equal amounts of protein and Streptavidin Sepharose (Cytiva) were incubated overnight at 4 °C by rotation. The beads were washed three times with lysis buffer and one time with PBS. Proteins were eluted from beads using SDS sample buffer supplemented with 2 mM biotin and then analyzed by western blotting.

## Protein purification

His-TEV-PRDX6 WT or C47S, His-TEV-hSCLY, hSEPHS2-His, mouse His-TEV-eEFSec and His-TEV-SBP2 were expressed in *E. coli* strain BL21-CodonPlus (DE3)-RIPL (Agilent Technologies). Expression of His-TEV-PRDX6 WT or C47S was induced by the addition of 0.2 mM IPTG, and culture was continued at 30 °C for 3 h. Cells were collected by centrifugation and frozen rapidly. Subsequently, cells were resuspended at 4 °C for 30 min in buffer containing 50 mM Tris-HCl (pH 8.0), 150 mM NaCl, 10 mM 2-mercaptoethanol (Nacalai Tesque), 200 µg ml⁻¹ lysozyme chloride (Nacalai Tesque), 10 µg ml⁻¹ DNase (Roche), 2 mM PMSF and a protease inhibitor cocktail (Roche), followed by lysis at 4 °C for 20 min in the presence of 0.2% Triton X-100. Insoluble material was removed by centrifugation at 23,700g for 20 min at 4 °C. His-TEV PRDX6 WT or C47S were purified from the supernatant using Ni-NTA beads (QIAGEN). The eluted samples were incubated for 30 min at 4 °C with 10 mM dithiothreitol (DTT) and then desalted in 20 mM Tris-HCl (pH 7.5) buffer on a PD10 column.

Expression of His-TEV-hSCLY and hSEPHS2-His was induced by the addition of 0.2 mM IPTG, and culture was continued overnight at 15 °C. Cells were collected by centrifugation and frozen rapidly. Subsequently, the cells were resuspended in buffer containing 50 mM Tris-HCl (pH 8.0), 150 mM NaCl, 10 mM 2-mercaptoethanol, 2 mM PMSF and a protease inhibitor cocktail and then lysed by sonication at 4 °C. Insoluble material was removed by centrifugation at 23,700g for 20 min at 4 °C. Proteins were purified using Ni-NTA beads. The eluted samples were desalted in 20 mM Tris-HCl (pH 7.5) and 1 mM DTT buffer on a PD10 column.

Expression of His-TEV-eEFSec and His-TEV-SBP2 was induced by the addition of 0.3 mM IPTG, and culture was continued at 30 °C for 4 h. Cells were collected by centrifugation and frozen rapidly. Subsequently, the cells were resuspended in buffer containing 20 mM Tris-HCl (pH 8.0), 500 mM NaCl, 10 mM imidazole, 10% glycerol, 10 mM 2-mercaptoethanol, 1 mM PMSF and a protease inhibitor cocktail and then lysed by sonication at 4 °C. Insoluble material was removed by centrifugation at 23,700g for 20 min at 4 °C. Proteins were purified using Ni-NTA beads. Eluted samples were desalted on a PD10 column. eEFsec was stored in buffer containing 20 mM Tris-HCl (pH 7.5), 150 mM NaCl and 1 mM DTT, and SBP2 was stored in buffer containing 10 mM Tris-HCl (pH 7.5), 150 mM NaCl, 5% glycerol and 1 mM DTT.

## GPX assay

GPX activity was measured using a GPX4 Inhibitor Screening Assay Kit (Cayman Chemical). First, 0.5 µg of GPX4 protein (included in the kit) or 5 µg of PRDX6 protein was dissolved in 50 µl of GPX4 assay buffer (included in the kit). Then, 20 µl of the GSH/GSH reductase mix (included in the kit) was added and mixed. Next, 20 µl of the NADPH solution (included in the kit) was added and mixed. Finally, 10 µl of cumene hydroperoxide (included in the kit), hydrogen peroxide (final concentration, 500 µM) or tert-butyl hydroperoxide (final concentration, 500 µM) was added and mixed, and absorbance

at 340 nm was measured using SpectraMax M5 microplate reader (Molecular Devices).

## Selenophosphate synthetase assay

A selenophosphate synthetase assay using SCLY and selenocysteine as a selenium source was performed in degassed buffer containing 50 mM Tris-HCl (pH 7.0), 100 µM ATP, 10 mM KCl, 10 mM MgCl$_2$, 1 mM DTT, 0.5 µM SEPHS2-His, 20 µM PRDX6 WT or C47S, 1 µM SCLY, 1 µM PLP (Sigma-Aldrich) and 50 µM (Sec)$_2$ at 30 °C for 45 min. The resultant AMP product was measured in an AMP-Glo assay (Promega) using a Nivo plate reader (PerkinElmer). A selenophosphate synthetase assay using selenide-bound PRDX6 as a selenium source was performed by incubating 200 µg of PRDX6 WT or C47S with 2 mM sodium selenite and 20 mM DTT in degassed 20 mM Tris-HCl (pH 7.5) buffer at room temperature for 30 min. Then, the solution was diluted 20 times in buffer containing 20 mM Tris-HCl (pH 7.5), 150 mM NaCl and 0.5% Triton X-100, and 30 µl of Ni-NTA magnetic beads was added. After rotation at 4 °C for 1 h, beads were washed three times and eluted with 300 mM imidazole in 20 mM Tris-HCl (pH 7.5) buffer. The eluted samples were incubated with a buffer mixture containing 50 mM Tris-HCl (pH 7.0), 100 µM ATP, 10 mM KCl, 10 mM MgCl$_2$, 1 mM DTT and 0.5 µM SEPHS2-His under a layer of mineral oil at 30 °C for 30 min. The resultant AMP product was measured in an AMP-Glo assay.

## RT–qPCR

RNA was isolated using the RNeasy Mini Kit (QIAGEN) and reverse transcribed using a high-capacity RNA-to-cDNA Kit (Applied Biosystems). RT–qPCR data were obtained by ABI ViiA7 Real-Time PCR system (Applied Biosystems) and analyzed by ViiA7 RUO Software v.1.2.3. RT–PCR of GPX4 was performed using the following primers:

mGPX4_Fwd, 5′-CCTCTGCTGCAAGAGCCTCCC-3′ and mGPX4_Rev, 5′-CTTATCCAGGCAGACCATGTGC-3′. Control primers were

β-mActin_Fwd, 5′-ATGGATGACGATATCGCTC-3′ and β-mActin_Rev, 5′-GATTCCATACCCAGGAAGG-3′.

## Isolation of aa-tRNAs

aa-tRNAs were isolated from WT, PRDX6 KO and SEPHS2 KO MEF cells. In brief, cells were treated (or not) with 50 nM (Sec)$_2$ for 105 min, followed by addition of 20 µg ml$^{-1}$ cycloheximide for 15 min. Cells were then washed three times in PBS and detached using trypsin. After centrifugation at 200g, 3.2 ml of DPEC water, 0.8 ml of 5× T buffer (50 mM NaOAc, 3.25 M NaCl, 50 mM MgCl$_2$ and 5 mM EDTA) and 2 ml of acid-phenol pH 4.2 (Nippon gene) were added to the cell pellets in that order. The solution was mixed briefly and centrifuged at 12,000g for 5 min at 4 °C. The aqueous phase was transferred to another tube and re-extracted with 2 ml of acid-phenol (pH 4.2). RNA was precipitated with 2.5 volumes of 100% ethanol, washed twice with 75% ethanol and air-dried for 5 min. The aa-tRNA pellet was resuspended in 5 mM NaOAc buffer.

## The luciferase reporter Sec UGA assay for Sec-tRNA[Ser]Sec evaluation

Each reaction (20 µl) contained wheat germ lysate (10 µl; Promega), 20 µM amino acid mix, 8 U of RNasin Plus ribonuclease inhibitor (Promega), 52.5 mM potassium acetate, 0.2 µl of luciferase reporter mRNA and 2.25 µg of aa-tRNAs in the presence or absence of 1.2 µg of eEFSec and 0.6 µg of SBP2. Reactions were incubated at 30 °C for 90 min. Luciferase activity was measured in a Lumat Luminometer (Berthold).

## ICP–MS

The reaction (100 µl) contained 40 µM sodium selenite, 160 µM GSH and 40 µM (or not) recombinant PRDX6 WT or C47S in degassed buffer containing 1 mM EDTA and 50 mM Tris-HCl (pH 7.0). Reactions were incubated at 25 °C for 5 min. Any selenium that did not bind to PRDX6 protein was removed using a NAP-5 column. The protein solution was then concentrated with an Amicon 3K concentrator. A small aliquot (50 µl) of the protein solution was put into a glass test tube, and the proteins were decomposed by addition of 0.1 ml of 60% HNO$_3$ followed by heating at 170°C for 2 h. The digested samples were diluted with Milli-Q water, and the concentration of selenium was measured by ICP-MS (Agilent 8800 ICP–MS/MS; Agilent Technologies). The O$_2$ mass-shift mode was used to monitor the signal intensity of $^{80}$Se$^{16}$O, with a dwell time of 100 ms. Selenium concentrations were measured from a standard calibration curve.

## Mass spectrometry analysis

The reaction contained recombinant PRDX6 WT (75 µM), sodium selenite (75 µM) and GSH (300 µM) in degassed buffer containing 50 mM Tris-HCl (pH 7.0), or PRDX6 WT (75 µM), Sec (75 µM) ((Sec)$_2$ was pre-reduced with an equimolar amount of TCEP), SCLY (1.5 µM) and PLP (1.5 µM) in degassed buffer containing 50 mM Tris-HCl (pH 7.0). Reactions were incubated at 25 °C for 5 min. Then, PRDX6 was alkylated with 10 mM iodoacetamide in 0.02% ProteaseMAX surfactant at 37 °C for 10 min, followed by digestion with 27 mg ml$^{-1}$ Trypsin Gold at 37 °C for 3 h. Products were analyzed using liquid chromatography–electrospray ionization–quadrupole time-of-flight tandem mass spectrometry (LC–ESI–Q-TOF MS/MS). LC–ESI–Q-TOF analysis was performed using a 6545XT AdvanceBio LC–Q-TOF apparatus (Agilent Technologies) connected to the Agilent HPLC system. Analysis of the modifications to the active center cysteine (Cys47) was performed using Agilent MassHunter BioConfirm software v.10.0. The modification levels in DFTPVCTTELGR peptide, which includes the active center cysteine residue, were detected by monitoring at $m/z$ 698.3341 (for CysS-AM (AM, iodoacetamide adduct)), $m/z$ 738.2942 (for CysSSe-AM) and $m/z$ 862.3172 (for CysSSe-SG). The efficiency of trypsin-mediated protein digestion was normalized using the FHDFLGDSWGILFSHPR peptide ($m/z$ 678.0041). The level of selenium modification in the fragment containing the active center cysteine was assessed by determining the relative ratio between the intensity of peptide-CysSSe-AM and that of peptide-CysS-AM. Samples of peptide-CysS-AM subjected to trypsin digestion followed by reduction with TCEP and subsequent alkylation with iodoacetamide were used to measure peptides corresponding to peptide-CysSSe-AM.

## Statical analysis and reproducibility

Data are presented as the mean ± s.d. or mean ± s.e.m. GraphPad Prism9 v.9.4.0 was used to calculate $P$ values using Student's $t$-test, one-way ANOVA or two-way ANOVA, followed by Tukey's multiple comparison test. The $P$ values are presented in the figure legends. All experiments were reproduced at least twice, each with similar results (the expression checks shown in Extended Data Figs. 1g and 3a were single experiments).

## Reporting summary

Further information on research design is available in the Nature Portfolio Reporting Summary linked to this article.

## Data availability

Boxplot data of differential gene expression in tumor and normal tissues were obtained from TNMplot (https://tnmplot.com/analysis). All data are available in the article and the supplementary information, and from the corresponding authors upon reasonable request. Source data are provided with this paper.

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

## Acknowledgements

We thank the members of the Iwai laboratory for helpful suggestions; Y. Saito (Tohoku University) and T. Toyama (Tohoku University) for kindly providing SELENOP; and F. Zhang (Massachusetts Institute of Technology) for providing the mouse GeCKO library and other CRISPR vectors. This work was supported by the Japan Society for the Promotion of Sciences (JSPS) KAKENHI: Grant Numbers 20H05505, 22H04810 and 22K06222 (to H.F.); by JSPS KAKENHI Grant Numbers 23K20040, 21H05263, 18H05277 and 22K19397 (to T.A.); by the Japan Science and Technology Agency CREST Grant Number JPMJCR2024 (to T.A.); by JSPS KAKENHI Grant Number 19H05772 (to Y.O.); and by JSPS KAKENHI Grant Number 24112002 and the Takeda Science Foundation (K.I.). The funders had no role in study design, data collection and analysis, decision to publish or preparation of the manuscript.

## Author contributions

H.F. and K.I. conceived and designed the project. H.F. performed most of the experiments. Y.T. performed ICP–MS analysis. S.O. performed mass spectrometry analysis. N.S., S.K., U.B., T.A. and Y.O. provided advice regarding experimental design. H.F. and K.I. wrote the manuscript, with contributions from all other authors.

## Competing interests

The authors declare no competing interests.

## Additional information

**Extended data** is available for this paper at https://doi.org/10.1038/s41594-024-01329-z.

**Correspondence and requests for materials** should be addressed to Hiroaki Fujita or Kazuhiro Iwai.

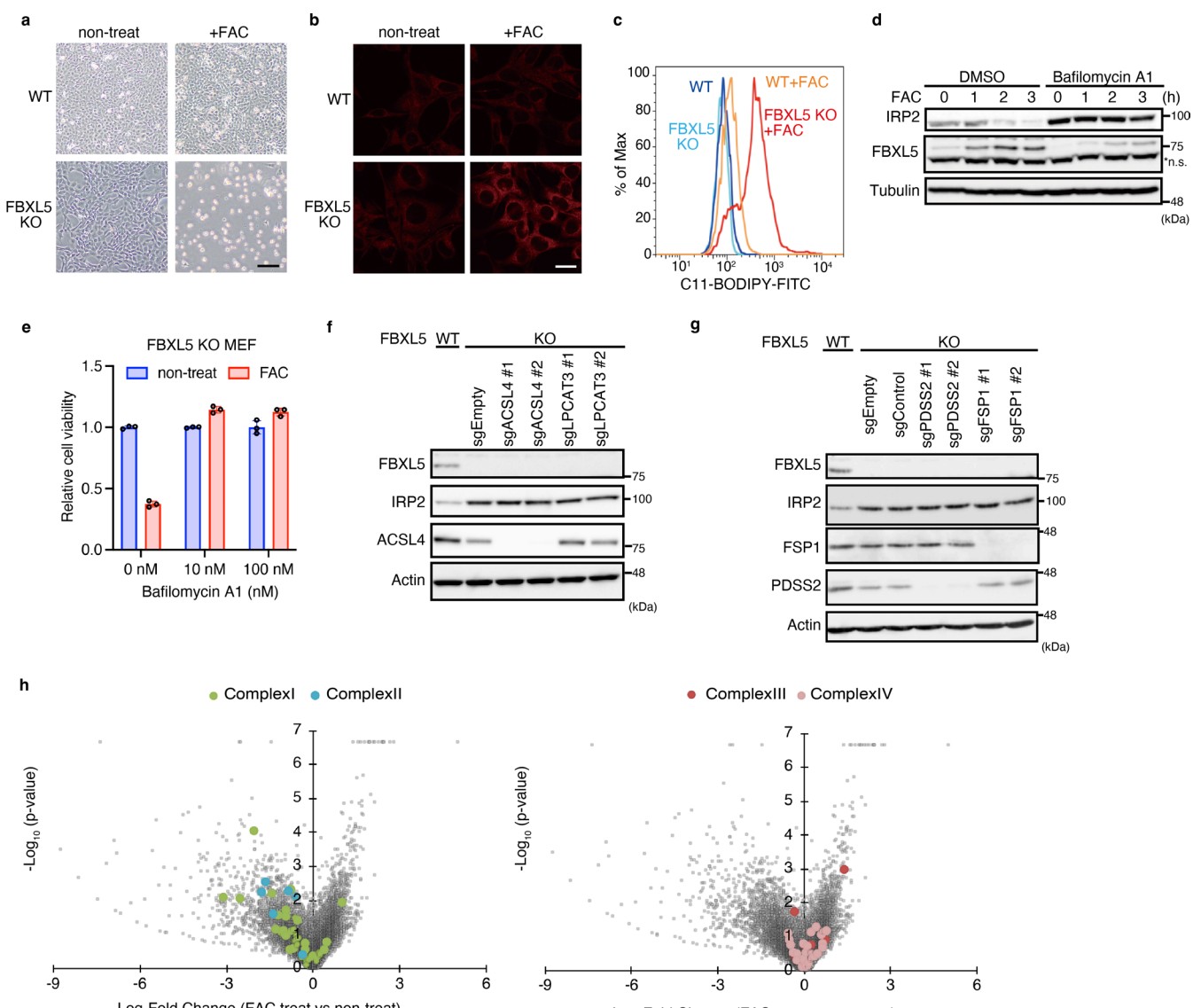

**Extended Data Fig. 1 | Characterization of iron-induced cell death in FBXL5 KO cells. a,** Phase-contrast images of cells after treatment with FAC (100 µg ml⁻¹) for 48 h. Scale bars, 200 µm. Data are representative of three independent experiments. **b,** Fluorescence microscopy images of cells stained with a FerroOrange probe after treatment for 4 h with FAC (25 µg ml⁻¹). Scale bars, 20 µm. Data are representative of three independent experiments. **c,** Flow cytometry analysis of lipid hydroperoxidation levels in WT or FBXL5 KO cells by C11-BODIPY staining after treatment with FAC (100 µg ml⁻¹l) for 24 h. Data are representative of two independent experiments. **d,** WT MEFs were pretreated for 3 h with bafilomycin A1 (100 nM), followed by FAC (100 µg ml⁻¹) for the indicated

times. Cell lysates were analyzed by immunoblotting. Data are representative of two independent experiments. **e,** Viability of FBXL5 KO MEFs treated for 24 h with FAC (100 µg ml⁻¹) in the presence of bafilomycin A1 (0, 10, 100 nM). Data are presented as the mean ± s.d. of three biological replicates. **f, g,** Immunoblot analysis of lysates from the indicated cells. Data (**f**) are representative of two independent experiments. Expression check of (**g**) is a single experiment. **h,** Volcano plots of iron-triggered ferroptosis screen, focusing on the mitochondrial electron transport chain (complex I and II: left, complex III and IV: right). The *p* value and fold change were generated by the MAGeCK test.

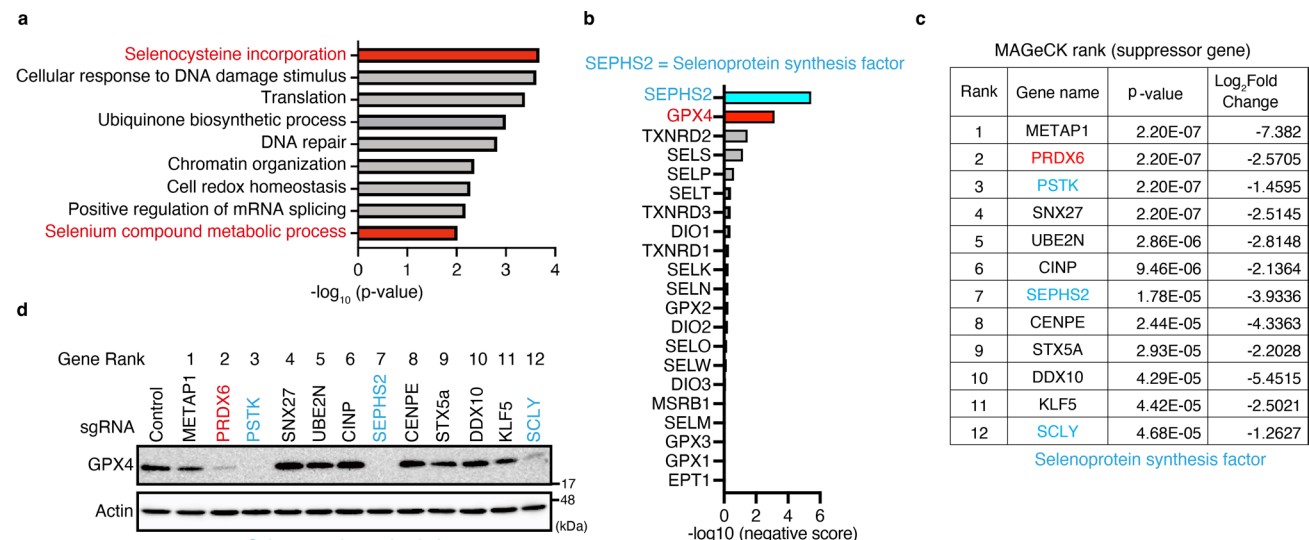

**Extended Data Fig. 2 | Validation of hit genes, and identification of PRDX6 as a GPX4 regulator. a**, Hit suppressor genes [p-value cut off (p < 0.001)] were examined by gene ontology analysis. The *p* value was generated by the David gene ontology test. **b**, Negative score ranking of selenoproteins from the screening list. **c**, MAGeCK rank of suppressor genes. The *p* value and fold change were generated by the MAGeCK test. **d**, Immunoblot analysis of lysates from cells expressing sgRNAs targeting the indicated genes. Data are representative of two independent experiments.

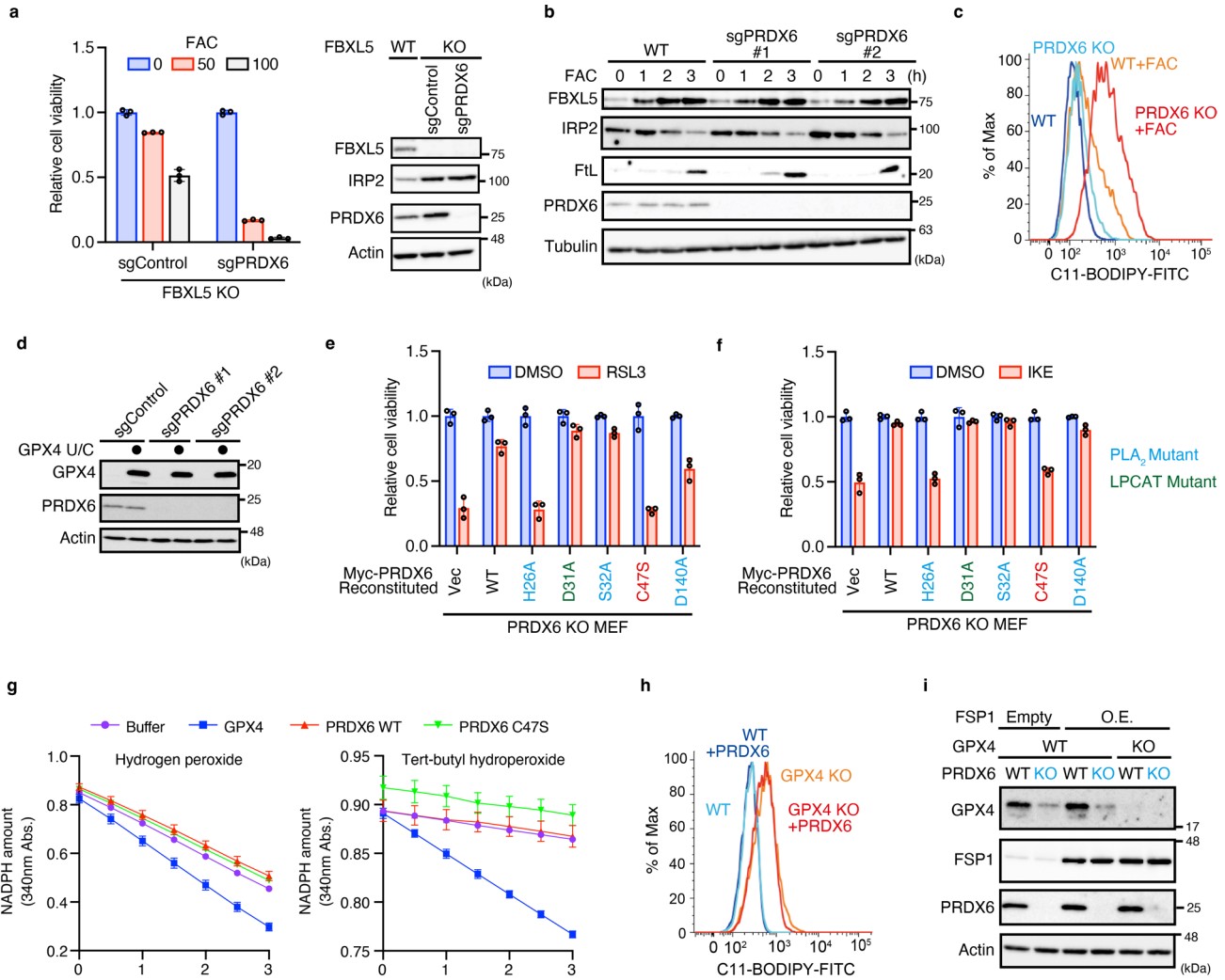

**Extended Data Fig. 3 | PRDX6 regulates ferroptosis by maintaining expression of GPX4. a**, Viability of FBXL5 KO cells or FBXL5/PRDX6 double knockout cells treated for 24 h with FAC (50 or 100 µg ml⁻¹), and immunoblot analysis of lysates from the indicated cells. Viability data are presented as the mean ± s.d. of three biological replicates. **b**, WT or PRDX6 KO MEFs were treated with FAC (25 µg ml⁻¹) for the indicated times, and cell lysates were analyzed by immunoblotting. Data are representative of two independent experiments. **c**, Flow cytometry analysis of lipid hydroperoxidation levels in WT and PRDX6 KO cells, as assessed by C11-BODIPY staining after treatment with FAC (50 µg ml⁻¹) for 90 min. Data are representative of three independent experiments. **d**, Immunoblot analysis of lysates from control of PRDX6 KO cells expressing the GPX4 U/C mutant. Data are representative of two independent

experiments. **e**, **f**, Viability of PRDX6 KO cells stably expressing the indicated PRDX6 mutants after incubation for 24 h in the presence of RSL3 (0.3 µM) (**e**) or IKE (0.3 µM) (**f**). Viability data are presented as the mean ± s.d. of three biological replicates. **g**, Measurement of GPX activity using recombinant proteins. Hydrogen peroxide (left) or tert-butyl hydroperoxide (right) were used as a substrate. Data are presented as the mean ± s.e.m of three independent experiments. **h**, Flow cytometry analysis of lipid hydroperoxidation after C11-BODIPY staining of WT or GPX4 KO cells stably expressing PRDX6 after removal of liiproxstatin-1 for 24 h. Data are representative of two independent experiments. **i**, Immunoblot analysis of lysates from the indicated cells. Data are representative of two independent experiments.

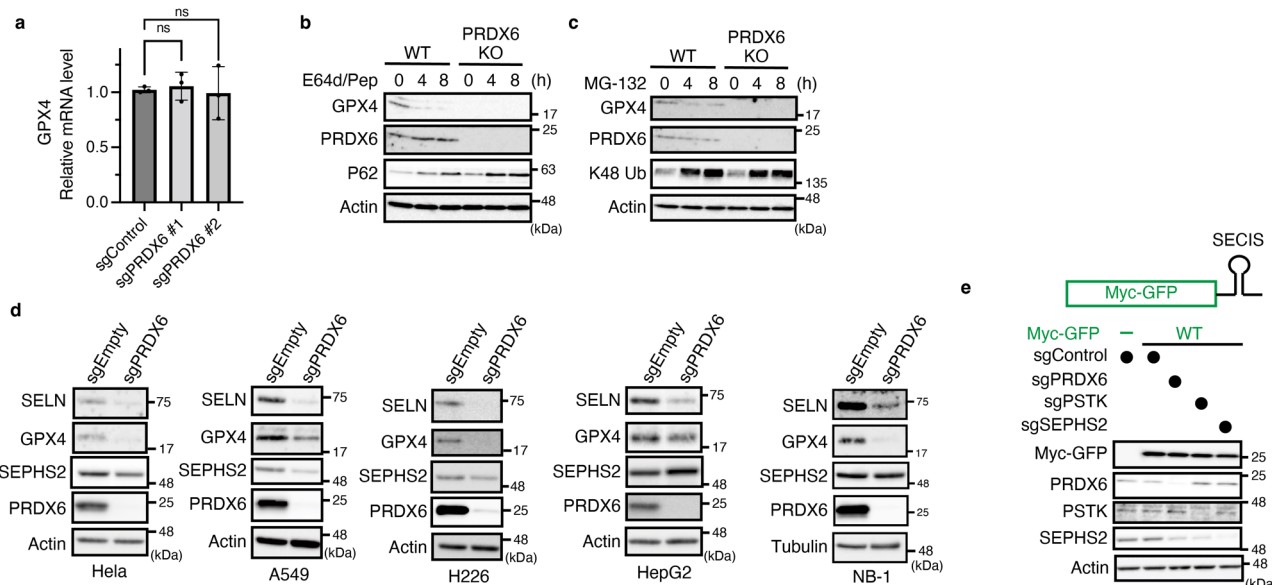

**Extended Data Fig. 4 | PRDX6 regulates expression of selenoproteins.**
**a**, Quantification of GPX4 mRNA levels by RT-qPCR. Data are expressed as the mean ± s.d of three biological replicates. ns; not significant; *P* (sgControl *vs*. sgPRDX6 #1 = 0.9523; sgControl *vs*. sgPRDX6 #2 = 0.9607); one-way ANOVA. **b**, **c**, Immunoblot analysis of lysates from control or PRDX6 KO cells incubated for 4 or 8 h in the presence of E64d/pep (10 µg ml⁻¹) (**b**) or MG-132 (10 µM) (**c**).

**d**, Immunoblot analysis of lysates from control or PRDX6 KO cells derived from the indicated human cancer cell lines (Hela, A549, H226, HepG2, NB-1). **e**, Immunoblot analysis of lysates from control cells or cells with KO of the indicated genes and stably expressing Myc-GFP WT. Data (**b-e**) are representative of two independent experiments.

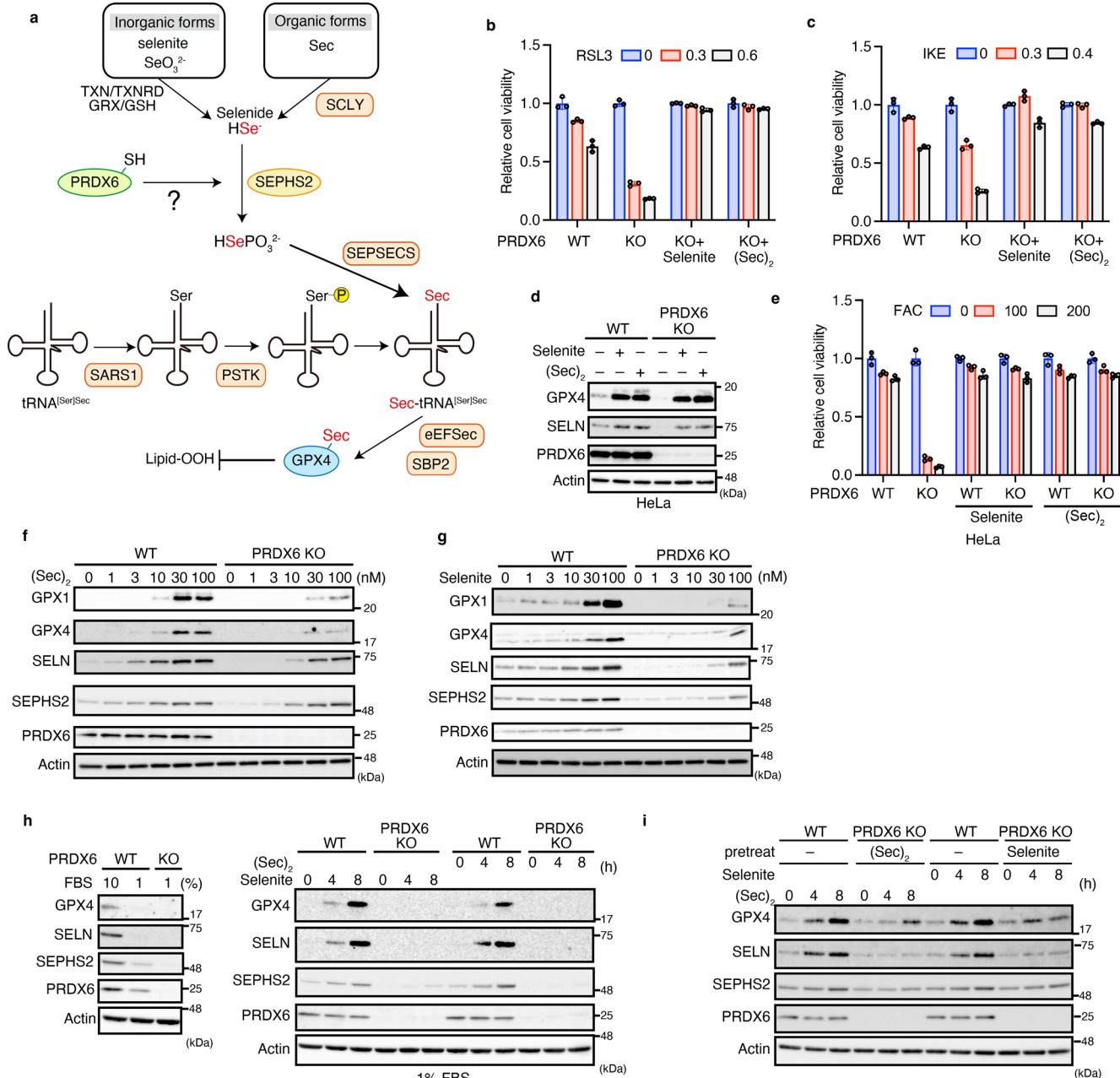

**Extended Data Fig. 5 | PRDX6 regulates selenium utilization efficiency.**
**a**, Schematic showing selenium metabolism and the selenoprotein synthesis pathway. **b**, **c**, WT or PRDX6 KO cells were pretreated for 2 days with (Sec)$_2$ (100 nM) or sodium selenite (100 nM). Then, cells were treated for 24 h with RSL3 (0.3 or 0.6 μM) (**b**) or IKE (0.3 or 0.4 μM) (**c**) in the presence of 100 nM (Sec)$_2$ or 100 nM sodium selenite. **d**, Immunoblot analysis of lysates from control or PRDX6 KO Hela cells cultured for 48 h in the presence of sodium selenite (100 nM) or (Sec)$_2$ (100 nM). **e**, WT or PRDX6 KO Hela cells were pretreated with (Sec)$_2$ (100 nM) or sodium selenite (100 nM) for 2 days. Then, cells were treated for 48 h with FAC (100 or 200 μg ml$^{-1}$) in the presence of (Sec)$_2$ (100 nM) or sodium selenite (100 nM), and cell viability was measured. **f**, **g**, Immunoblot analysis of

lysates from control or PRDX6 KO cells cultured for 32 h in the presence of the indicated concentrations of (Sec)$_2$ (**f**) or sodium selenite (**g**). **h**, WT or PRDX6 KO cells were incubated for 24 h with 1% FBS in the presence of liproxstatin-1. Then, cells were treated with 50 nM (Sec)$_2$ or 100 nM sodium selenite for the indicated times. Cell lysates were analyzed by immunoblotting. **i**, PRDX6 KO cells were preincubated for 36 h with 10 nM (Sec)$_2$ or 40 nM sodium selenite. Then, WT or PRDX6 KO cells pretreated with selenium were exposed to 50 nM (Sec)$_2$ or 200 nM sodium selenite for the indicated periods and the lysates were analyzed by immunoblotting. Data are representative of two (**d**, **h**, **i**) or three (**f**, **g**) independent experiments. Viability data (**b**, **c**, **e**) are presented as the mean ± s.d. of three biological replicates.

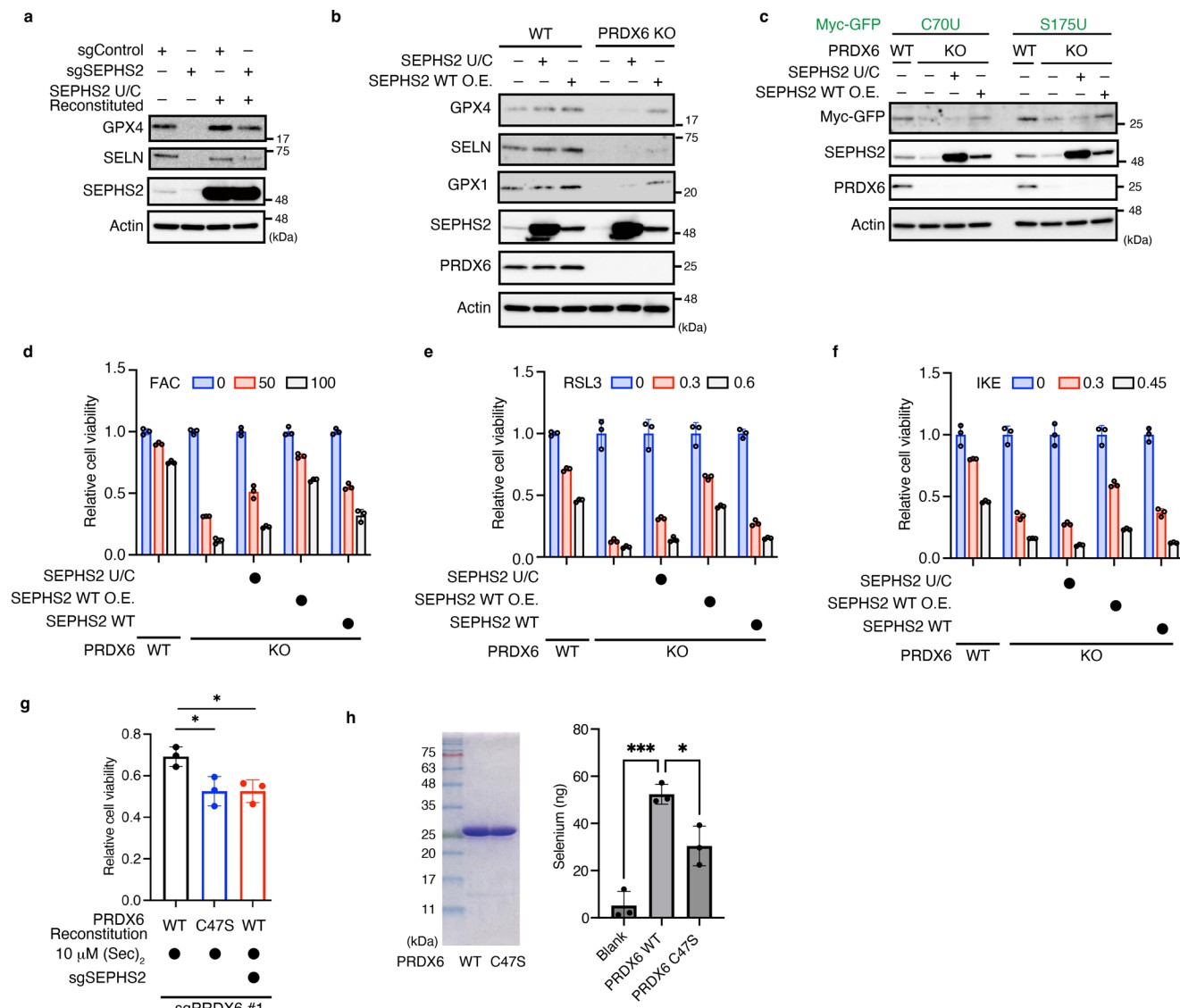

**Extended Data Fig. 6 | Overexpression of SEPHS2 WT rescues the PRDX6 KO phenotype. a**, Immunoblot analysis of lysates from control or SEPHS2 KO cells stably expressing the SEPHS2 U/C mutant. **b**, Immunoblot analysis of lysates from control or PRDX6 KO cells stably expressing the SEPHS2 WT or U/C mutant. **c**, Immunoblot analysis of lysates from cells stably expressing Myc-GFP C70U or S175U in control or PRDX6 KO cells stably expressing the SEPHS2 WT or U/C mutant. Data (**a-c**) are representative of two independent experiments. **d-f**, Viability of the indicated cells in the presence of FAC (50 or 100 μg ml⁻¹) for 48 h (**d**), or RSL3 (0.3 or 0.6 μM) (**e**) or IKE (0.3 or 0.45 μM) (**f**) for 24 h. Viability

data are presented as the mean ± s.d. of three biological replicates. **g**, Viability of the indicated cells treated for 48 h with (Sec)₂ (10 μM) and liproxstatin-1 (3 μM). Viability data are presented as the mean ± s.d. of three biological replicates. *P (PRDX6 WT *vs*. C47S = 0.0243; WT *vs*. sgSEPHS2 = 0.0238) < 0.05; one-way ANOVA. **h**, Coomassie-stained SDS-PAGE gel showing the recombinant PRDX6 WT or C47S mutant used for ICP-MS analysis. ICP-MS analysis of the amount of selenium bound to recombinant PRDX6 WT or C47S. Data are presented as the mean ± s.d. of three independent experiments. *P (=0.0141) < 0.05, ***P (=0.0003) < 0.001; one-way ANOVA.

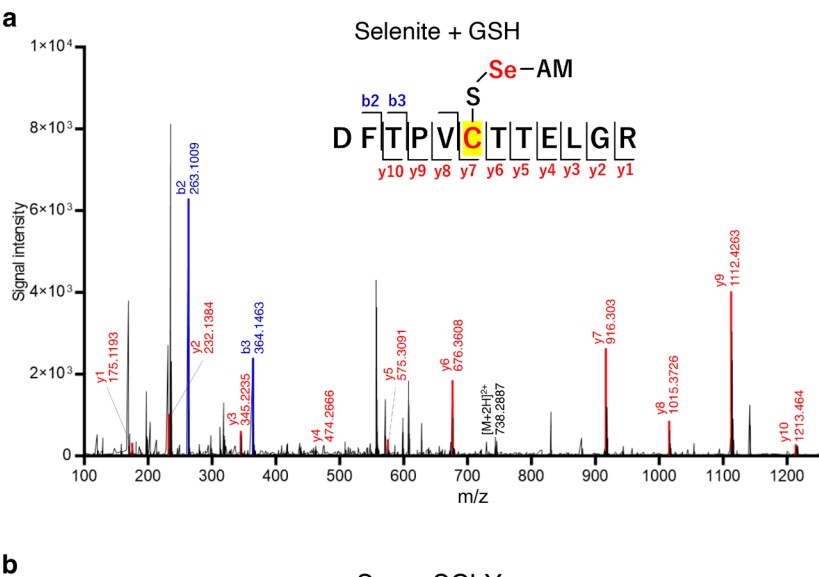

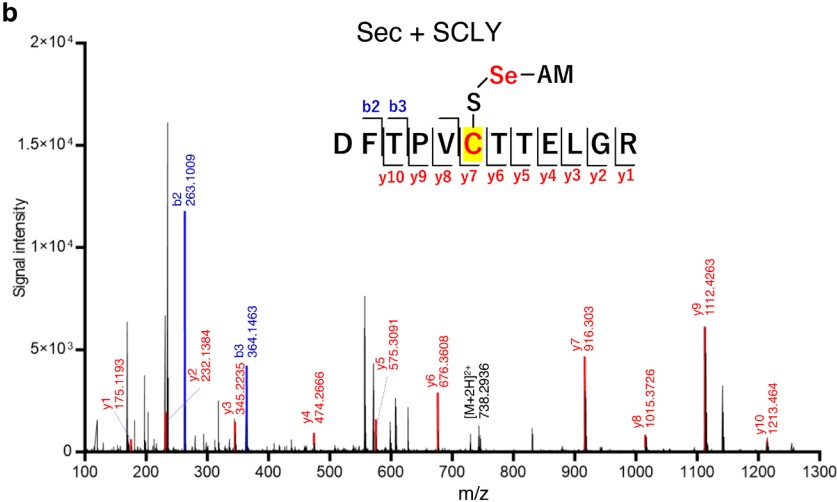

**Extended Data Fig. 7 | Mass spectrometry analysis of PRDX6 C47 modification. a, b,** Mass spectrum of PRDX6 C47. PRDX6 (75 µM) was incubated with GSH (300 µM) and sodium selenite (75 µM) (**a**), or with Sec (75 µM) and SCLY (1.5 µM) (**b**) for 5 min prior to mass spectrometry analyses.

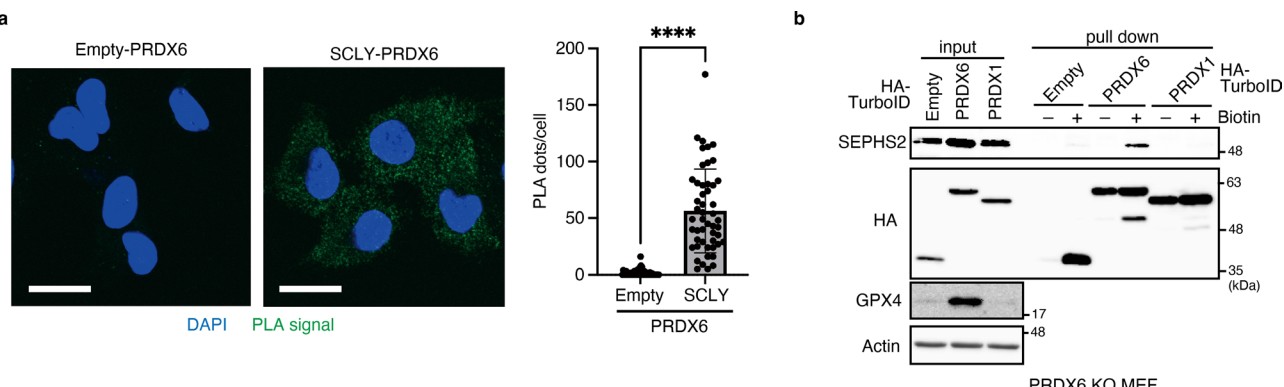

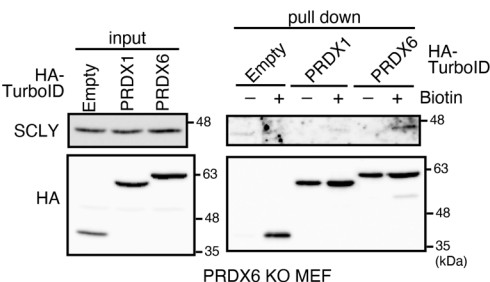

**Extended Data Fig. 8 | Interaction between PRDX6 and SEPHS2 or SCLY.**
**a**, The physical association between PRDX6 and SCLY was detected in a PLA assay. Scale bars, 20 μm. Data are representative of three independent experiments. Quantification of the PLA dots is also shown. Data are presented as the mean ± s.d. [n = 52 cells (Empty-PRDX6), n = 50 cells (SCLY-PRDX6) examined], ****$P < 0.0001$;

unpaired two-sided t-test. **b**, **c** PRDX6 KO MEFs expressing HA-TurboID-empty, -PRDX1, or -PRDX6 were cultured for 30 min with DMSO or 50 μM biotin. Cell lysates and pulldown samples were analyzed by immunoblotting. The association between PRDX6 and SEPHS2 (**b**) or between PRDX6 and SCLY (**c**) was analyzed. Data are representative of two (**c**) or three (**b**) independent experiments.

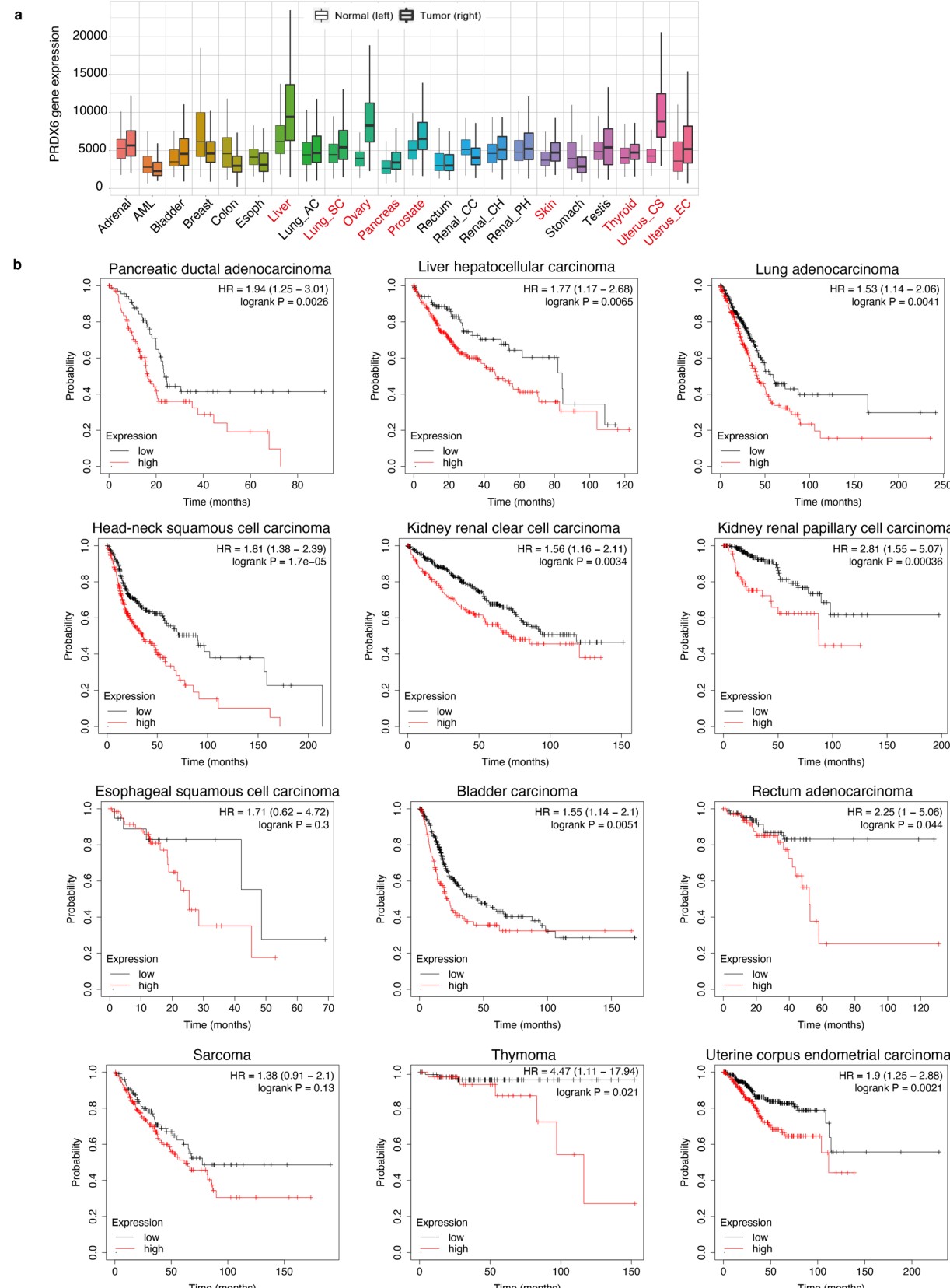

**Extended Data Fig. 9 | Expression of PRDX6 is associated with a poor prognosis for several cancers. a**, Expression profile of PRDX6 in several types of normal and tumor tissue. Data were obtained from TNMplot (https://tnmplot. com/analysis/), which is a web server that stores data regarding expression of normal and cancer-related genes. Those with higher values in tumor tissue compared to normal tissue and with significant differences (Mann-Whitney U test) are marked in red (*P* < 0.01). **b**, Kaplan-Meier survival curves comparing cancer samples showing high PRDX6 expression with samples showing low PRDX6 expression (curves were constructed using Kaplan-Meier plotter).

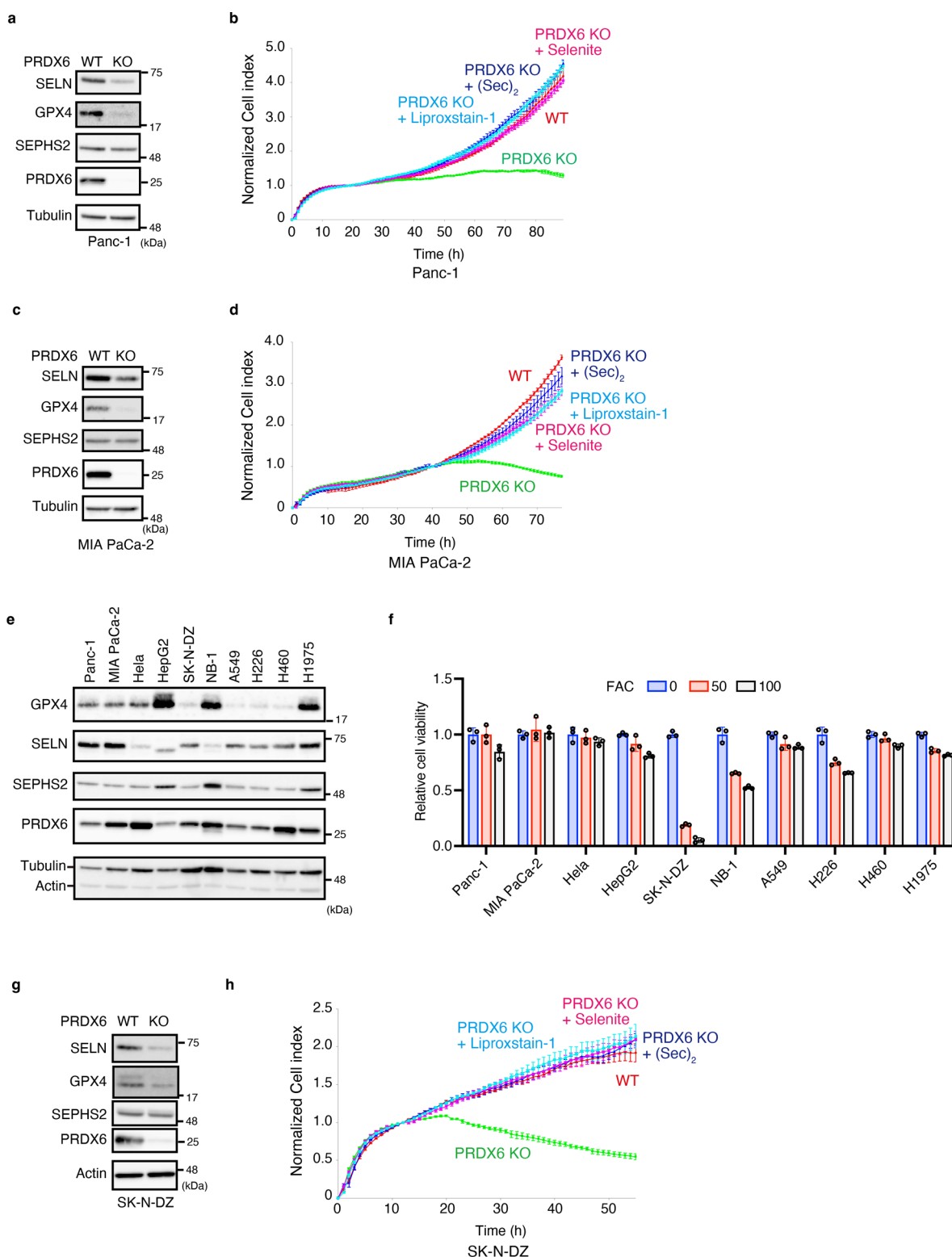

**Extended Data Fig. 10 | Expression of PRDX6 is important for survival of cancer cell lines. a**, Immunoblot analysis of lysates from WT or PRDX6 KO Panc-1 cells. **b**, Continuous monitoring of the viability of WT or PRDX6 KO Panc-1 cells in the presence or absence of sodium selenite (100 nM), (Sec)$_2$ (100 nM), or liproxstatin-1 (1 μM). **c**, Immunoblot analysis of lysates from WT or PRDX6 KO MIA PaCa-2 cells. **d**, Continuous monitoring of the viability of WT or PRDX6 KO MIA PaCa-2 cells in the presence or absence of sodium selenite (100 nM), (Sec)$_2$ (100 nM), or liproxstatin-1 (1 μM). **e**, Immunoblot analysis of lysates from indicated cell lines. **f**, Viability of indicated cell lines in the presence of FAC (50 or 100 μg ml$^{-1}$) for 48 h. Viability data are presented as the mean ± s.d. of three biological replicates. **g**, Immunoblot analysis of lysates from WT or PRDX6 KO SK-N-DZ cells. **h**, Continuous monitoring of the viability of WT or PRDX6 KO SK-N-DZ cells in the presence or absence of sodium selenite (100 nM), (Sec)$_2$ (100 nM), or liproxstatin-1 (1 μM). Viability data (**b, d, h**) are presented as the mean ± s.e.m of three biological replicates. Data (**a, c, e, g**) are representative of two independent experiments.

# Reporting Summary

## Statistics

For all statistical analyses, confirm that the following items are present in the figure legend, table legend, main text, or Methods section.

| n/a | Confirmed | |
|---|---|---|
| ☐ | ☒ | The exact sample size (*n*) for each experimental group/condition, given as a discrete number and unit of measurement |
| ☐ | ☒ | A statement on whether measurements were taken from distinct samples or whether the same sample was measured repeatedly |
| ☐ | ☒ | The statistical test(s) used AND whether they are one- or two-sided<br>*Only common tests should be described solely by name; describe more complex techniques in the Methods section.* |
| ☒ | ☐ | A description of all covariates tested |
| ☐ | ☒ | A description of any assumptions or corrections, such as tests of normality and adjustment for multiple comparisons |
| ☐ | ☒ | A full description of the statistical parameters including central tendency (e.g. means) or other basic estimates (e.g. regression coefficient) AND variation (e.g. standard deviation) or associated estimates of uncertainty (e.g. confidence intervals) |
| ☐ | ☒ | For null hypothesis testing, the test statistic (e.g. *F*, *t*, *r*) with confidence intervals, effect sizes, degrees of freedom and *P* value noted<br>*Give P values as exact values whenever suitable.* |
| ☒ | ☐ | For Bayesian analysis, information on the choice of priors and Markov chain Monte Carlo settings |
| ☒ | ☐ | For hierarchical and complex designs, identification of the appropriate level for tests and full reporting of outcomes |
| ☒ | ☐ | Estimates of effect sizes (e.g. Cohen's *d*, Pearson's *r*), indicating how they were calculated |

*Our web collection on statistics for biologists contains articles on many of the points above.*

## Software and code

Policy information about availability of computer code

| | |
|---|---|
| Data collection | Western blotting anaylysis was measured by a LAS4000mini or LAS3000 instrument (GE Healthcare).Cell viability assay was measured by SpectraMax M5(Molecular Device). Immunofluorescence images were collected using Fv1000 microscopy (Olympus). DNA and RNA concentrations were measured by Nanodrop2000 (Thermo).Protein concentration was measured by SpectraMax M5(molecular device) or V750 spectrophotometer (JASCO). GPX assay (absorbance at 340 nm) was measured by SpectraMax M5 (Molecular Devices).RT-qPCR data were obtained by ABI ViiA7 Real-Time PCR system (Applied Biosystems). Luciferase reporter was measured by a Lumat Luminometer (Berthold). The AMP product was measured by plate reader Nivo (PerkinElmer). Cell survival data on iCelligence system were acquired by using RTCA iCelligence Software (ACEA Biosciences). ICP-MS was done by Agilent 8800 ICP-MS/MS (Agilent Technologies). LC-ESI-Q-TOF analysis was performed using 6545XT AdvanceBio LC/Q-TOF (Agilent Technologies) connected to the Agilent HPLC system. Flow cytometry data was collected by using FACS Canto II (BD Biosciences) and FACS Diva software ver 6.1.2 (Becton Dickinson). |
| Data analysis | GraphPad Prism9 version 9.4.0 was used to perform statistical analysis. CRISPR screen data was analyzed by MAGeCK pipeline. Microsoft Exel ver16.72 was used for data analysis. Image Gauge ver4.22 (FUJIFILM) was used for chemiluminescent images. ViiA7 RUO Software ver1.2.3 (Thermo Fisher Scientific) was used for RNA expression analysis. Recording data of cell survival using iCelligence system were analyzed by RTCA Software Lite version2.2.1(ACEA Biosciences). Immunofluorescence images were analyzed by ImageJ (ver2.3.0). FlowJo software (Tomy Digital Biology, version 9.9.6) was used for all analyses of flow cytometry data. The modification analysis of the active center cysteine (Cys47) was performed using MassHunter BioConfirm software ver10.0 (Agilent Technologies). Figures in the manuscript were arranged and converted into PDF files by using illustrator version 27.4.1 (Adobe). |

For manuscripts utilizing custom algorithms or software that are central to the research but not yet described in published literature, software must be made available to editors and reviewers. We strongly encourage code deposition in a community repository (e.g. GitHub). See the Nature Portfolio guidelines for submitting code & software for further information.

## Data

Policy information about availability of data

All manuscripts must include a data availability statement. This statement should provide the following information, where applicable:
- Accession codes, unique identifiers, or web links for publicly available datasets
- A description of any restrictions on data availability
- For clinical datasets or third party data, please ensure that the statement adheres to our policy

> Data of differential gene expression in tumor and normal tissues were obtained from TNMplot (https://tnmplot.com/analysis/). All data are available in the article and the supplementary information, and from the corresponding authors upon reasonable request.

## Human research participants

Policy information about studies involving human research participants and Sex and Gender in Research.

| | |
|---|---|
| Reporting on sex and gender | N/a |
| Population characteristics | N/a |
| Recruitment | N/a |
| Ethics oversight | N/a |

Note that full information on the approval of the study protocol must also be provided in the manuscript.

# Field-specific reporting

Please select the one below that is the best fit for your research. If you are not sure, read the appropriate sections before making your selection.

☒ Life sciences       ☐ Behavioural & social sciences       ☐ Ecological, evolutionary & environmental sciences

For a reference copy of the document with all sections, see nature.com/documents/nr-reporting-summary-flat.pdf

# Life sciences study design

All studies must disclose on these points even when the disclosure is negative.

| | |
|---|---|
| Sample size | No statistical tests were used to determine sample size. Sample size was determined on the previous studies in the field using similar experiment paradigms [PMID: 34031600 and 31634899],  and to give sufficient values to conduct standard statistical tests. |
| Data exclusions | No data were excluded from the analyses. |
| Replication | All experiments were reproduced at least twice, each with similar results (the expression checks shown in extended data Figures 1g and 3a were single experiments). |
| Randomization | Not applicable as no animal studies were performed. |
| Blinding | Blinding was not done because the investigator needs to know the treatment groups in order to perform the experiments. |

# Reporting for specific materials, systems and methods

We require information from authors about some types of materials, experimental systems and methods used in many studies. Here, indicate whether each material, system or method listed is relevant to your study. If you are not sure if a list item applies to your research, read the appropriate section before selecting a response.

## Materials & experimental systems

| n/a | Involved in the study |
|---|---|
| ☐ | ☒ Antibodies |
| ☐ | ☒ Eukaryotic cell lines |
| ☒ | ☐ Palaeontology and archaeology |
| ☒ | ☐ Animals and other organisms |
| ☒ | ☐ Clinical data |
| ☒ | ☐ Dual use research of concern |

## Methods

| n/a | Involved in the study |
|---|---|
| ☒ | ☐ ChIP-seq |
| ☐ | ☒ Flow cytometry |
| ☒ | ☐ MRI-based neuroimaging |

## Antibodies

| | |
|---|---|
| Antibodies used | anti-PRDX6 (Proteintech, 13585-1-AP; western blotting [WB], 1:2,000; Proximity ligation assay [PLA], 1:100), anti-SEPHS2 (Proteintech, 14109-1-AP; WB, 1:2,000), anti-GPX4 (Proteintech, 67763-1-AP; WB, 1:2,000), anti-FSP1 (Proteintech, 20886; WB, 1:2000), anti-SCLY (Proteintech, 67606-1-Ig; WB, 1:2000), anti-GPX4 (Santa Cruz, sc-166570; WB, 1:2,000), anti-SELN (Santa Cruz, sc-365824; WB, 1:2,000), anti-GPX1/2 (Santa Cruz, sc-133160; WB, 1:2,000), anti-PSTK (Santa Cruz, sc-373991; WB, 1:2,000), anti-FBXL5 (Santa Cruz, sc-390102; WB, 1:2,000), anti-FTH1 (Santa Cruz, sc-376594; WB, 1:300), anti-ACSL4 (Santa Cruz, sc-365230; WB, 1:2000), anti-PDSS2 (Santa Cruz, sc-515137; WB, 1:2000), anti-ferritin (Sigma-Aldrich, F6136; WB, 1:2,000), anti-IRP2 (in-house; WB, 1:1000), anti-LRP8 (abcam, ab108208; WB, 1:2000), anti-b-Actin (Sigma-Aldrich, A5316; WB, 1:15,000), anti-Tubulin (CEDARLANE, CLT9002; WB, 1:5,000), anti-Myc (Merck, 05-724; WB, 1:2,000; PLA, 1:200), anti-HA (MBL, M180-3; WB, 1:2000), anti-Ubiquitin K48-specific (Merck, ZRB2150; WB, 1:2000), anti-p62 (Wako, 018-22141; WB, 1:2,000), HRP-linked antimouse IgG (Cell Signaling, #7076; WB, 1:10,000), and HRP-linked antirabbit IgG (GE Healthcare, NA934; WB, 1:10,000). |
| Validation | Primary antibodies were validated in this study or used in previous articles for WB of mouse or human sample.<br><br>Anti-PRDX6 (Proteintech, 13585-1-AP), anti-SEPHS2 (Proteintech, 14109-1-AP), Anti-GPX4 (Proteintech, 67763-1-AP) and (Santa Cruz, sc-166570), anti-GPX1/2 (Santa Cruz, sc-133160), anti-PSTK (Santa Cruz, sc-373991), anti-FBXL5 (Santa Cruz, sc-390102), anti-ACSL4 (Santa Cruz, sc-365230), anti-PDSS2 (Santa Cruz, sc-515137), have been validated by knockout of the endogenous genes with CRISPR/Cas9.<br>Anti-FTH1 (Santa Cruz, sc-376594), anti-ferritin (Sigma-Aldrich, F6136) and Anti-IRP2 (in-house) have been validated in previous publication (PMID: 35318808)<br>Anti-FSP1 (Proteintech, 20886) has been validated in previous publication (PMID: 31634900).<br>Anti-SELN (Santa Cruz, sc-365824) and anti-LRP8 (abcam, ab108208) have been validated in previous publication (PMID: 35637349).<br><br>Validation detail for the other commercial antibodies is on the manufacture's website that is attached as following.<br>Anti-SCLY (Proteintech, 67606-1-Ig);https://www.ptglab.com/products/SCLY-Antibody-67606-1-Ig.htm; and has been also validated by expression of tagged-SCLY construct.<br>Anti-b-Actin (Sigma-Aldrich, A5316); https://www.sigmaaldrich.com/JP/en/product/sigma/a5316<br>Anti-Tubulin (CEDARLANE, CLT9002); https://www.cedarlanelabs.com/Products/Detail/CLT9002<br>Anti-Myc (Merck, 05-724); https://www.merckmillipore.com/JP/en/product/Anti-Myc-Tag-Antibody-clone-4A6,MM_NF-05-724<br>Anti-HA (MBL, M180-3); https://ruo.mbl.co.jp/bio/e/dtl/A/?pcd=M180-3<br>Anti-Ubiquitin K48-specific (Merck, ZRB2150); https://www.sigmaaldrich.com/JP/en/product/sigma/zrb2150<br>Anti-p62 (Wako, 018-22141); https://labchem-wako.fujifilm.com/us/product/detail/W01W0101-2214.html<br>HRP-linked antimouse IgG (Cell Signaling, #7076); https://www.cellsignal.com/products/secondary-antibodies/anti-mouse-igg-hrp-linked-antibody/7076?country=JP&language=en<br>HRP-linked antirabbit IgG (GE Healthcare, NA934); https://www.cytivalifesciences.com/en/us/shop/protein-analysis/blotting-and-detection/blotting-standards-and-reagents/amersham-ecl-hrp-conjugated-antibodies-p-06260 |

## Eukaryotic cell lines

Policy information about cell lines and Sex and Gender in Research

| | |
|---|---|
| Cell line source(s) | Mouse embryonic fibroblasts (MEFs) were generated in-house. HepG2 was gifted by Dr. Koichi Nakajima (Osaka City University), originally purchased from ATCC (HepG2 : HB-8065). HEK293T was gifted by Dr. Eijiro Nakamura (Kyoto University), originally purchased from RIKRN RBC (293T : RCB2202). PLATE was gifted by Dr. Toshio Kitamura (Tokyo University). A549, H226, H460 and H1975 were gifted by Dr. Atsuyasu Sato (Kyoto University), originally purchased from ATCC (A549 : CCL-185, H226 : CRL-5826, H460 : HTB-177, H1975 : CRL-5908). SK-N-DZ and HeLa cells were purchased from ATCC (SK-N-DZ : CRL-2149, Hela : CCL-2). PANC-1 and MIA Paca-2 cells were purchased from RIKEN RBC (PANC-1 : RCB2095, MIA Paca-2 : RCB2094). NB-1 cells were purchased from JCRB (NB-1 : JCRB0621). |
| Authentication | None of the cell lines used were not authenticated. |
| Mycoplasma contamination | All cell lines tested negative for mycoplasma contamination. |
| Commonly misidentified lines<br>(See ICLAC register) | No commonly misidentified cell line was used. |

# Flow Cytometry

## Plots

Confirm that:

☒ The axis labels state the marker and fluorochrome used (e.g. CD4-FITC).

☒ The axis scales are clearly visible. Include numbers along axes only for bottom left plot of group (a 'group' is an analysis of identical markers).

☒ All plots are contour plots with outliers or pseudocolor plots.

☒ A numerical value for number of cells or percentage (with statistics) is provided.

## Methodology

| | |
|---|---|
| Sample preparation | Cells were plated in a 6-well plate and treated with FAC. After 1.5–24 h, the medium was removed, and the cells were labeled with DMEM containing 10 uM BODIPY 581/591 C11 at 37°C for 30 min. Cells were then washed three times with PBS and detached from the plate using trypsin, and green fluorescence was measured by flow cytometry using a BD FACSCanto II cytometer |
| Instrument | FACSCanto II (BD Biosciences) for flow cytometric analyses |
| Software | FACS Diva software ver 6.1.2 (Becton Dickinson) for data collection. FlowJo version 9.9.6 for data analysis. |
| Cell population abundance | At least 5000 cells were analyzed for each sample. |
| Gating strategy | FSC-area vs SSC-area was used to gate for the bulk populations of cells. Initial cell population gating (FSC-Area vs FSC-height) was adopted to make sure doublet exclusion and only single cell was used for analysis. |

☒ Tick this box to confirm that a figure exemplifying the gating strategy is provided in the Supplementary Information.

