## [Peer Review File · Nature Structural & Molecular Biology]

Peer Review Information

Manuscript Title: PRDX6 augments selenium utilization to limit iron toxicity and ferroptosis

Corresponding author name(s): Kazuhiro Iwai, Hiroaki Fujita

Reviewer Comments & Decisions:

Decision Letter, initial version:
--

Message: 25th Jul 2023

Dear Dr. Iwai,

Thank you again for submitting your manuscript "PRDX6 augments selenium utilization to limit iron toxicity and ferroptosis". I apologise for the delay in reaching back with a decision, which resulted from the difficulty in obtaining referee reports in a timely fashion. The expert whose report was delayed has asked us to convey their sincere apologies. Nevertheless, we now have comments (below) from the 3 reviewers who evaluated your paper. In light of those reports, we remain interested in your study and would like to see your response to the comments of the referees, in the form of a revised manuscript.

You will see that all authors appreciate the potential of the study in providing crucial insight in selenium transfer and its interplay with ferroptosis. However, all experts bring up important issues which require addressing in a revised manuscript. More specifically, you will note that both reviewer #1 and reviewer #3 (points 3-6) put forward a series of suggestions to adequately substantiate the mechanistic and functional findings in this manuscript, including, but not limited to, exemplifying the functional interaction between PRDX6 and SEPHS2, characterising the PRDX6-selenium intermediate, and identifying the preferred source of selenium (organic, inorganic) by PRDX6. Along similar lines, reviewer #2 requests including a ferroptosis inhibitor in the selenium rescue experiments of PRDX6-KO cells and reviewer #3 wants to see some of the functional data about the potential relevance of the PRDX6/GPX4 axis expanded (point 2). Finally, all authors require clarifications and resolving textual or experimental ambiguities which need to be resolved.

Please be sure to address/respond to all concerns of the referees in full in a point-by-point response and highlight all changes in the revised manuscript text file. If you have comments that are intended for editors only, please include those in a separate cover letter.

We are committed to providing a fair and constructive peer-review process. Do not

hesitate to contact us if there are specific requests from the reviewers that you believe are technically impossible or unlikely to yield a meaningful outcome.

We expect to see your revised manuscript within 3-6 months. If you cannot send it within this time, please contact us to discuss an extension; we would still consider your revision, provided that no similar work has been accepted for publication at NSMB or published elsewhere.

Reporting Summary:

When submitting the revised version of your manuscript, please pay close attention to our [href="https://www.nature.com/nature-portfolio/editorial-policies/image-integrity">Digital Image Integrity Guidelines](https://www.nature.com/nature-portfolio/editorial-policies/image-integrity). and to the following points below:

Data availability: this journal strongly supports public availability of data. All data used in accepted papers should be available via a public data repository, or alternatively, as

Supplementary Information. If data can only be shared on request, please explain why in your Data Availability Statement, and also in the correspondence with your editor. Please note that for some data types, deposition in a public repository is mandatory - more information on our data deposition policies and available repositories can be found below: <https://www.nature.com/nature-research/editorial-policies/reporting-standards#availability-of-data>

Nature Structural & Molecular Biology is committed to improving transparency in authorship. As part of our efforts in this direction, we are now requesting that all authors identified as 'corresponding author' on published papers create and link their Open Researcher and Contributor Identifier (ORCID) with their account on the Manuscript Tracking System (MTS), prior to acceptance. This applies to primary research papers only. ORCID helps the scientific community achieve unambiguous attribution of all scholarly contributions. You can create and link your ORCID from the home page of the MTS by clicking on 'Modify my Springer Nature account'. For more information please visit please visit www.springernature.com/orcid.

[Redacted]

Sincerely,

Dimitris Typas
Associate Editor
Nature Structural & Molecular Biology
ORCID: 0000-0002-8737-1319

Referee expertise:

Referee #1: ferroptosis, GPX4, cell death pathways, selenium metabolism-cell death interplay

Referee #2: ferroptosis, selenium metabolism-cell death interplay

Referee #3: Molecular biology of ferroptosis and cancer, CRISPR screens+ferroptosis

Reviewers' Comments:

Reviewer #1:

Remarks to the Author:

This paper presents a potentially groundbreaking investigation into mechanisms of selenium transfer. The current work proposes a novel function for PRDX6 in enhancing selenium utilization to mitigate iron toxicity and ferroptosis. The authors propose that PRDX6 acts as a unique selenoprotein biosynthesis factor, indirectly exerting antioxidant effects through selenoprotein expression, thereby challenging its previously presumed function as a GPX-like enzyme. While previous studies have associated PRDX6 with ferroptosis (PMID: 31036877, 35258176, and 32778843), this study unveils a "non-canonical" mechanism, introducing exciting and intriguing findings. It is worth noting that a related work not mentioned by the authors (PMID - 36008942) hints at a potential role for PRDX6 in selenocysteine metabolism, although it does not explicitly demonstrate it. Despite this, the research holds promise and despite currently being preliminary it could be of substantial interest to the scientific community if further supported with additional evidence for this novel function.

Therefore I believe the points highlighted below are essential and reasonable in order to provide the sound conceptual advance expected from such work.

Main points requiring attention:

- In their model system, how does the loss of PRDX6 impact the sensitivity to canonical ferroptosis inducers? Analyzing the PRDX6 isogenic pairs against known ferroptosis inducers could be valuable to understand if the process is only limited to iron-mediated cell death or general ferroptosis regulation.
- The authors exclude a GPX-like function of recombinant PRDX6 though this is not properly addressed. In the materials and methods there is a lack of adequate information about the substrate used to monitor GPX activity. To validate the activity against different peroxides, such as hydrogen peroxide and organic peroxides, like t-BOOH and phospholipid hydroperoxides, would be crucial given the proposed new function.
- The interaction between PRDX6 and SEPHS2, while plausible, is not adequately demonstrated. The PLA assay used may result in false positive signals, possibly arising from highly abundant proteins like PRDX6. While PLAs can provide information about molecular interactions occurring in close proximity, they do not offer high-resolution details about the exact spatial arrangement or distances between molecules. The sensitivity and accuracy of proximity ligation assays are heavily dependent on the quality and specificity of the primary antibodies used to detect the target molecules. If the antibodies used have low specificity, they may cross-react with other proteins, resulting in non-specific signals. A more reliable assay, such as FRET or validation of a direct interaction through crosslinking-IP, is necessary to support the proposed transfer mechanism. Similar concerns are also related to the data shown in Extended Data Fig 4

for SCLY.

- The selenium binding to PRDX6 alone is insufficient evidence for the proposed mechanism. As PRDX6 has two cysteines, mutating the catalytic C47 to S reduces the amount of selenium bound, but 50% continues to be bound to the protein (definitely not trace amounts as the authors claim), suggesting unspecific binding of selenide/selenite to cysteine. This could be likely the case for any PRDX or any cysteine that is reactive/exposed enough. Therefore further support for this claim is needed, especially considering this might be a general reaction of selenite with any available cysteine. Mass-spectrometry analysis is necessary to characterize the identity of this selenium-PRDX6 adduct. Moreover, a relative quantification of this adduct (modified against unmodified PRDX6) needs to be demonstrated in order to provide insights into how efficient this reaction is. For example, if only 1% of the protein is modified after 24h, it is unlikely that this would be a relevant transfer system in a cellular context.
- Following on the point above, it is critically important that the authors demonstrate the identity of the PRDX6-Selenium intermediate. It is proposed that the only available selenium form that can be transferred to SEPHS2 is via a perselenide intermediate. A direct PRDX6-selenite adduct is unlikely to carry such a transfer. Therefore, it is of fundamental importance to verify whether this intermediate is indeed generated in the reaction of PRDX6 with selenite.
- The SEPHS2 enzymatic assays presented in Figure 4m and n lack convincing evidence, as the signal-to-noise ratio is extremely low, and substantial AMP conversion appears to occur even without the enzyme. Ideally, the author should provide the results as mols of ADP and not as relative values. In line with this, a better characterization of the assay and direct proof of selenophosphate generation would be critical to unequivocally support these findings (though this could be challenging). A good starting point would be to benchmark this with the original Ogasawara/Stadtman work (PMID 11493708), where very clear differences can be observed.
- Results presented in Figure 4, where the authors feed selenium in both organic and inorganic forms (selenite and selenocysteine), argue against the proposed mechanism since PRDX6 KO cells promptly metabolize these intermediates to generate new selenoproteins and hint at an alternative form. The data does not support the model depicted in Extended Figure 4 where all selenide would travel via PRDX6. While the assays addressing time and concentration-dependent effects are well taken, their interpretation is not straightforward because the PRDX6-KO cells start with an already lower level of expression. An important aspect would be to consider using radiolabeled selenium sources, as this would be the only way to unambiguously identify de novo selenoprotein biosynthesis. Additionally, the authors might wish to consider a more physiological source of selenium, for example, SELENOP.

Additional points:

- The authors' validation for the knockouts used in the studies is insufficiently provided, as seen in Figure 1E, F: Supplementary 2c. Given the relevance of this information in determining if the FBXL5-KO dies via ferroptosis in the presence of iron, it is essential to conduct proper characterization.
- The quality of some of the western blots, like Figure 3c, needs improvement. Although the authors claim that the reconstitution of PRDX6-WT fully rescues selenoprotein translation, the exposure of GPX1 and GPX4 appears to be extremely low. To ensure a proper comparison, higher exposure should be considered for such experiments.
- Correct lipid peroxides to lipid hydroperoxides
- The PLA experiments seem to be done in A549 cells, unlike the rest of the experiments, which are performed in MEF's. The authors need to explain why this experiment was done

in another system. The A549 cell line is known to have a mutation in KEAP1 so the “antioxidant” context might be completely different. A validation that loss of PRDX6 in these cells phenocopies the previous results could suffice to put this point to rest.
- Fig. 1f. please mark the position of PDSS2 on the volcano plot in Fig. 1d

Reviewer #2:

Remarks to the Author:

The paper “PRDX6 augments selenium utilization to limit iron toxicity and ferroptosis” reports peroxiredoxin6 (PRDX6) as a novel component involved in selenoprotein synthesis. Through a genome-wide CRISPR screen for iron-triggered ferroptosis and subsequent gene ontology analysis of the identified ferroptosis suppressors, selenoprotein synthesis factors were identified. Moreover, experimentation on the identified ferroptosis suppressors revealed that the deletion of PRDX6 decreased GPX4 expression. Findings from additional experiments suggested that PRDX6 indirectly suppresses ferroptosis by augmenting GPX4 activity. Because selenoprotein synthesis factors were identified as a top hit from the initial screening and further analysis from database tools suggested a connection between PRDX6 and the selenoprotein synthetic pathway, PRDX6 was hypothesized to be a novel key player involved in selenoprotein synthesis. Further results supported the involvement of PRDX6 in the selenoprotein synthetic pathway and efforts to probe the specific mechanism at play suggested that PRDX6 is a selenide carrier protein that helps facilitate Sec-tRNA[Ser]Sec synthesis. PRDX6 was also evaluated in the context of cancer, in which database analyses revealed higher PRDX6 expression in cancerous tissues compared to normal tissues. In pancreatic cancer cell lines, PRDX6 KO reduced cell viability, while the addition of selenium sources restored viability. Taken together, the identification of PRDX6 as an intracellular selenium carrier involved in the selenoprotein synthesis pathway not only offers novel mechanistic insight, but also suggests the targeting of PRDX6 in this pathway as a potential therapeutic strategy in cancers where high PRDX6 expression is correlated with poor clinical outcomes.

Strengths

- The systematic nature in which PRDX6 was first identified and then probed, made this paper easy to follow.
- The experiments conducted to probe the mechanism by which PRDX6 is involved in the selenoprotein synthetic pathway are thorough and diverse, ranging from CRISPR screening, database analyses, and downstream assays. The in vitro reconstructive system for selenoprotein synthesis developed by the authors was noteworthy.
- The figures are high quality.
- In addition to dissecting the mechanism in which PRDX6 is involved in selenoprotein synthesis, the authors studied PRDX6 in the context of cancers. This offered more insight into the application of their findings and suggested the targeting of selenoprotein synthesis machinery as a therapeutic strategy in these cancer types.

Weaknesses

Major

- Line 54
o The authors report that “IRP2 stabilization caused by loss of FBXL5 led to accumulation of redox active iron in cells” Figure 1A suggests that IRP1 may be involved too. Is it possible to say with certainty that IRP2 stabilization led to this? What about IRP1?
- Line 79

o At this point, it might be too early to say that “the loss of selenoprotein synthesis factors seem to sensitize cells to iron-triggered ferroptosis by reducing GPX4.” This would fit better after bringing up Figure 2C.

- Line 135-136

o The loss of PRDX6 reducing selenoprotein expression across many human cancer cell lines does not suggest that the role of PRDX6 was conserved across species, that would entail including non-human groups as well. Instead, the authors might want to consider re-framing that to suggesting that the role of PRDX6 is conserved across many cancer types.

- Line 227-228

o The authors claim that “inhibition of PRDX6 may provide an effective strategy for sensitizing cancer cells to iron-triggered ferroptosis”
 o However, the data suggest that viability is decreased under PRDX6 KO but is rescued with the addition of selenium. Including a ferroptosis inhibitor in these experiments would support the claim that PRDX6 inhibition might sensitize the cancer cells to ferroptosis. As it stands, there is no ferroptosis-specific data in this experiment to make such claim.

Minor

- Line 37

o Expand GPX4. This is the first time it is mentioned in the main body of the text.

- Line 40

o Expand FBXL5 for the same reason.

- Typo in Line 47-line 48:

o “.....revealed (remove extra that) increased peroxidation in iron-repleted FBXL5 KO cells”

- Line 51

o It would be helpful to explicitly discuss how the unbiased genome-wide CRISPR screening was performed on the FBXL5 KO cells that were treated with FAC or untreated. While this is captured in the figure, it would make it much more clear if explicitly said in the text as well.

- Figure 1B

o It would be helpful to include the concentrations of liproxstatin in the image itself.

- Line 86

o It might be beneficial to elaborate more on why PRDX6 did not affect iron homeostasis.

- Line 106

o Missing the word “a” after “however” and before “controversial”

- Line 132

o This is the first time encountering SEPHS2 in the main body of the text- please expand in the first mention.

- Line 139

o Expand PSTK- this is the first mention.

- Line 154
 - o Typo. "Finally, we explored the role of PRDX6 in selenoprotein synthesis (or in the selenoprotein synthetic pathway)."
- Line 191-193
 - o This could benefit from some restructuring.
- Line 197
 - o "...and then transfer"
- Line 198
 - o "...revealed that endogenous PRDX6 is effectively.."

Reviewer #3:

Remarks to the Author:

In this manuscript, the authors identified PRDX6 as a novel intracellular selenium carrier. They provided impressive evidence to show that PRDX6 augmented the intracellular selenium utilization efficiency as a selenium carrier and promoted selenoprotein synthesis in general. The function of PRDX6 in ferroptosis is somewhat controversial in the previous studies. This study provides a clear mechanism that loss of PRDX6 promotes ferroptosis indirectly decreasing GPX4 abundance. Overall, this study provides novel viewpoint to the field of selenoprotein and ferroptosis. To improve the current version and make the conclusion more convincing, the authors need to conduct additional experiments and address several important points as following:

1. The authors performed a secondary screen using the abundance of GPX4 as a readout with sgRNA against a small panel of genes (extended data Fig2C). This screen identified PRDX6 as a negative regulator for GPX4 protein level. However, it is not described how they narrowed down their screen to this list of genes. The authors should provide the rational here.
2. PRDX6 is important for GPX4 translation, and its expression is associated with poor prognosis of several cancers. To improve the significance of this study, the authors can collect a panel of cancer cells (e.g., PDAC) and compare the expression level of PRDX6, GPX4, SEPHS2 in these cell lines, as well as to compare their sensitivity to iron or RSL3-induced ferroptosis.
3. Unlike other selenoproteins, SEPHS2 is somehow less affected by PRDX6 knockout (fig 3b, 3c, extended Fig 3d, etc.). Knockout of PRDX6 in HepG2 cells even significantly increased SEPHS2 expression. Sine SEPHS2 is responsible for phosphorylating hydroxyselenide to selenophosphate, it is possible that SEPHS2 itself can directly bind to selenide without the help of PRDX6 (although the affinity might be much weaker). The authors tested that SEPHS2 U/C mutant did not rescue PRDX6 KO phenotype, however, they should also test whether SEPHS2 wt overexpression can rescue PRDX6 KO phenotype. In Fig 3D and Fig 4J, they should also test the conditions of PRDX6ko + SEPHS2 wt or SEPHS2 mutant to rule out this possibility.
4. To confirm that PRDX6 suppresses ferroptosis through maintaining GPX4 expression, the authors need to test whether overexpression of PRDX6 wt and mutants (as in Fig2D)

can suppress ferroptosis triggered by RSL3 and IKE, in addition to FAC. More importantly, the authors should establish a cellular system with genotype of GPX4-KO/FSP1-overexpression as in PMID 37267948 (these GPX4-null cells can grow normally because of FSP1 overexpression). Based on the mechanism proposed by the authors, in these cells PRDX6 won't be able to suppress IKE-induced ferroptosis.

5. The difference of the selenophosphate synthetase assay readout in Fig 4m, 4n is rather marginal. More convincing results are needed here (hopefully this can be achieved by optimizing assay condition?). The SEPHS2-His used in this experiment is purified from bacteria. Guess it is the U-to-C mutant? The authors might need to purify SEPHS2 wt using a eucaryotic expression system.

Minor points:

1. In extended data Fig3d, knockout of PRDX6 in HepG2 cells significantly increased SEPHS2 expression but only slightly decreased GPX4 expression. Why is that, and does PRDX6 knockout affect ferroptosis sensitivity in HepG2?

2. The authors need to show SEPHS2 expression in Fig 4D, 4E.

3. Why is PRDX6 not an essential gene in most of the cancer cell lines, which is very different to other core components of selenoprotein translation pathways (as shown by DepMap)?

4. Please describe PLA assay in more detail in the method part.

Author Rebuttal to Initial comments

Common statements to all reviewers

We sincerely thank all three reviewers for their insightful and helpful comments. In accordance with these comments, we performed a range of experiments which have improved the depth of our study substantially. We are pleased to inform the reviewers that we have now been able to add the following key results:

- (1) We performed various experiments in which we treated cells with canonical ferroptosis inducers. The data clearly showed that PRDX6 plays crucial roles in protecting not only iron-induced ferroptosis but also ferroptosis triggered by canonical ferroptosis inducers.
- (2) We found that overexpression of SEPHS2 WT substantially, albeit not completely, restored expression of some selenoproteins in PRDX6-deficient cells; however, SEPHS2 failed to restore selenoprotein expression when expressed at the endogenous level. These results suggest that SEPHS2 can utilize selenium without help from PRDX6, albeit less efficiently, and reinforce our conclusion that PRDX6 plays a role in efficient utilization of selenium by SEPHS2. Also, this observation supports our findings that excess selenium can induce expression of selenoproteins in PRDX6 KO cells. To include these results in the revised text, we have reordered the data in previous Figures 3 and 4.

- (3) Mass spectrometry analysis identified the S-Se bond of PRDX6 C47.
- (4) Binding of PRDX6 and SEPHS2, shown by the PLA assay, was also demonstrated in a TurboID experiment.

In addition, we revised the text in accordance with the reviewer's suggestions. We also reorganized the manuscript to better fit the journal's style.

We sincerely hope that the reviewers find our revised study suitable for publication in *Nature Structural & Molecular Biology*.

Our point-by-point responses to the reviewers' comments are attached, as well as further details regarding the experiments performed and conclusions drawn from them.

Our replies to the reviewers' comments are shown in blue, while the reviewers' comments are shown in *black italics*. All changes in the manuscript are highlighted in yellow.

Response to Reviewer #1

This paper presents a potentially groundbreaking investigation into mechanisms of selenium transfer. The current work proposes a novel function for PRDX6 in enhancing selenium utilization to mitigate iron toxicity and ferroptosis. The authors propose that PRDX6 acts as a unique selenoprotein biosynthesis factor, indirectly exerting antioxidant effects through selenoprotein expression, thereby challenging its previously presumed function as a GPX-like enzyme. While previous studies have associated PRDX6 with ferroptosis (PMID: 31036877, 35258176, and 32778843), this study unveils a "non-canonical" mechanism, introducing exciting and intriguing findings. It is worth noting that a related work not mentioned by the authors (PMID - 36008942) hints at a potential role for PRDX6 in selenocysteine metabolism, although it does not explicitly demonstrate it. Despite this, the research holds promise and despite currently being preliminary it could be of substantial interest to the scientific community if further supported with additional evidence for this novel function.

Therefore I believe the points highlighted below are essential and reasonable in order to provide the sound conceptual advance expected from such work.

Response: We thank the reviewer for the constructive comments. We believe that the new data have improved our study markedly. We have addressed each of the comments below.

Main points requiring attention:

–In their model system, how does the loss of PRDX6 impact the sensitivity to canonical ferroptosis inducers? Analyzing the PRDX6 isogenic pairs against known ferroptosis inducers could be valuable to understand if the process is only limited to iron-mediated cell death or general ferroptosis regulation.

Response: We appreciate the suggestion to expand the study to include an examination of the crucial role of PRDX6-mediated selenium transfer in protection from ferroptosis in general. As suggested, we performed several additional experiments. First, we treated PRDX6 KO cells with different gRNAs in the presence of RSL3 and the erastin-derivative imidazole ketone erastin (IKE) (Fig. 2c,d). We also examined whether expression of PRDX6 mutants (Extended Data Fig. 3e,f), addition of selenium (Extended Data Fig. 5b,c), and overexpression SEPHS2 (Extended Data Fig. 6e,f) protect cells from ferroptosis induced by RSL3 and IKE. The results clearly show that the mechanisms identified in the original study also play a role in general ferroptosis.

–The authors exclude a GPX-like function of recombinant PRDX6 though this is not properly addressed. In the materials and methods there is a lack of adequate information about the substrate used to monitor GPX activity. To validate the activity against different peroxides, such as hydrogen peroxide and organic peroxides, like t-BOOH and phospholipid hydroperoxides, would be crucial given the proposed new function.

Response: We apologize for providing insufficient information with respect to monitoring the GPX activity of PRDX6 in the Materials and methods section. We have now provided a full and detailed description of the experiment, and performed a new experiment using hydrogen peroxide or tert-butyl hydroperoxide as a substrate (Fig. 2g and Extended Data Fig. 3g). We also performed the following experiment, although we decided not to include it the manuscript. Considering that Glutathione-S-Transferase (GSTP1) was shown to enhance GPX activity of PRDX6¹, we examined whether purified GSTP1 augments the GPX activity of PRDX6, and found that it did not (Reviewer Fig. 1). Many previous reports detected the GPX activity of PRDX6 in PRDX6-knockdown or -KO cells²⁻⁴; however, we believe that this led to the misunderstanding that PRDX6 has GPX activity because expression of GPX proteins such as GPX1 and GPX4 falls markedly upon loss of PRDX6.

Reviewer Figure 1.

(a) Measurement of GPX activity using recombinant proteins. GPX4 (0.5 µg), PRDX6 (5 µg), PRDX6 (5 µg) + GSTP1 (5 µg).

–The interaction between PRDX6 and SEPHS2, while plausible, is not adequately demonstrated. The PLA assay used may result in false positive signals, possibly arising from highly abundant proteins like PRDX6. While PLAs can provide information about molecular interactions occurring in close proximity, they do not offer high-resolution details about the exact spatial arrangement or

distances between molecules. The sensitivity and accuracy of proximity ligation assays are heavily dependent on the quality and specificity of the primary antibodies used to detect the target molecules. If the antibodies used have low specificity, they may cross-react with other proteins, resulting in non-specific signals. A more reliable assay, such as FRET or validation of a direct interaction through crosslinking-IP, is necessary to support the proposed transfer mechanism. Similar concerns are also related to the data shown in Extended Data Fig 4 for SCLY.

Response: Thank you. We performed TurboID experiments to confirm binding of PRDX6 to SEPHS2. We utilized the TurboID system, which labels proximal proteins with biotin, because PRDX6 is thought to dissociate from SEPHS2 upon transfer of selenium, which implies that the interaction between the two proteins is weak and transient. Therefore, we expressed TurboID-PRDX6 or TurboID-PRDX1 (a paralog of PRDX6 as a control) in WT and PRDX6 KO MEFs to show that PRDX6 binds specifically to SEPHS2 (Fig. 5g and Extended Data Fig. 8b). We also showed that PRDX6 binds to SCLY using an identical system (Extended Data Fig. 8c).

The selenium binding to PRDX6 alone is insufficient evidence for the proposed mechanism. As PRDX6 has two cysteines, mutating the catalytic C47 to S reduces the amount of selenium bound, but 50% continues to be bound to the protein (definitely not trace amounts as the authors claim), suggesting unspecific binding of selenide/selenite to cysteine. This could be likely the case for any PRDX or any cysteine that is reactive/exposed enough. Therefore further support for this claim is needed, especially considering this might be a general reaction of selenite with any available cysteine. Mass-spectrometry analysis is necessary to characterize the identity of this selenium-PRDX6 adduct. Moreover, a relative quantification of this adduct (modified against unmodified PRDX6) needs to be demonstrated in order to provide insights into how efficient this reaction is. For example, if only 1% of the protein is modified after 24h, it is unlikely that this would be a relevant transfer system in a cellular context.

Response: We thank the reviewer for this critical comment. To consolidate our conclusions, we performed mass-spectrometry analyses and successfully detected a perselenide (-S-Se) intermediate on C47 of PRDX6 when we incubated purified PRDX6 with selenite+GSH or with SCLY+Sec (Extended Data Fig. 7). We believe that this is the first time that the S-Se bond has been identified on the Cys residue of proteins by mass spectrometry. We found that about 10% of C47 was modified with S-Se when PRDX6 reacted with GSH+selenite or SCLY+Sec, and that this occurred in as little as 5 minutes (Fig. 5d,e). It was reported previously that when C47 of PRDX6 is modified, PRDX6 forms a multimer⁵; thus, we suspect that all C47 is not modified with S-Se, possibly because C47 of PRDX6 is masked after formation of multimers by PRDX6. Alternatively, it is possible that the S-Se bond is unstable and was lost during the analyses. We believe that assessment of these possibilities is beyond the scope of this paper, but we would like to examine them in the future.

- Following on the point above, it is critically important that the authors demonstrate the identity of the PRDX6-Selenium intermediate. It is proposed that the only available selenium form that can be transferred to SEPHS2 is via a perselenide intermediate. A direct PRDX6-selenite adduct is unlikely to carry such a transfer. Therefore, it is of fundamental importance to verify whether this intermediate is indeed generated in the reaction of PRDX6 with selenite.

Response: As noted in the reply to the above comment, we detected a perselenide intermediate on PRDX6 C47. Furthermore, ICP-MS analysis revealed that GSH is critical for addition of Se to PRDX6 when PRDX6 reacts with selenite (Reviewer Fig. 2). Since GSH is abundant in cells, we suspect that GSH within cells facilitates formation of a perselenide intermediate on PRDX6.

Reviewer Figure 2.

(a) PRDX6 (40 μ M) was incubated for 5 min with selenite (40 μ M) or selenite (40 μ M) +GSH (160 μ M). Then, excess selenium was removed on a Nap-5 column, and PRDX6-bound selenium was analyzed by ICP-MS.

- The SEPHS2 enzymatic assays presented in Figure 4m and n lack convincing evidence, as the signal-to-noise ratio is extremely low, and substantial AMP conversion appears to occur even without the enzyme. Ideally, the author should provide the results as mols of ADP and not as relative values. In line with this, a better characterization of the assay and direct proof of selenophosphate generation would be critical to unequivocally support these findings (though this could be challenging). A good starting point would be to benchmark this with the original Ogasawara/Stadtman work (PMID 11493708), where very clear differences can be observed.

Response: We thank the reviewer for the critical comment. As reviewer points out, we realized that SEPHS2 U/C constitutively degrades ATP to AMP, even in the absence of selenide, as reported previously^{6,7}. Therefore, the increase in SEPHS2 U/C-mediated AMP production upon addition of PRDX6 WT was not large, although it was statistically significant (Fig. 5h,i). To improve the experimental system, we purified SEPHS2 U/C using two tags (GST and MBP) that are different His₆ tag; however, SEPHS2 U/C also degraded ATP to AMP in the absence of selenium, regardless of the tags (Reviewer Fig. 3a). It might be possible that degradation of ATP to AMP in the absence of selenide is lower in SEPHS2 WT than in SEPHS2 U/C. Therefore, we tried to purify SEPHS2 WT. Although a previous report showed that SEPHS2 WT can be purified using a baculovirus expression system, the authors used a construct lacking SECIS within the 3'-UTR⁸. To purify SEPHS2 containing Sec, we generated a recombinant baculovirus

containing a SEPHS2 WT construct with a 3'-UTR containing SECIS; however, we could not obtain enough SEPHS2 WT using this system. We then decided to purify SEPHS2 WT from cells overexpressing SEPHS2 WT, although it was poorly expressed (Fig. 5a). Unfortunately, purified SEPHS2 WT also constitutively degraded ATP to AMP, although addition of selenium increased the generation of AMP (Reviewer Fig. 3b). Finally, we measured selenophosphate synthesis by SEPHS2 directly in NMR experiments. Selenophosphate was detected by NMR using inorganic or organic selenium as a selenium source; however, there was no significant increase in selenophosphate upon addition of PRDX6 (Reviewer Fig. 4a–c). We found that PRDX6 plays a role the efficiency of selenium utilization on a nanomolar scale (Fig. 4d, e and Extended Data Fig. 5f–i). and that SEPHS2 utilizes selenium under conditions of high selenium concentrations, even in the absence of PRDX6 (Fig. 4a–c). We had to add much higher concentrations of selenium (millimolar) to the experiment due to the poor sensitivity of NMR. At such high concentrations, it is likely that selenium would be transferred to SEPHS2 in the absence of PRDX6. Although we tried to improve the experimental procedure to increase the efficacy and detect the selenophosphate synthesis activity of SEPHS2, we were unsuccessful (possibly due to degradation of ATP to AMP by SEPHS2 in the absence of selenium). Thus, we believe that our current results, which show with statistical significance that PRDX6 increases production of AMP by SEPHS2 is the best result that we can provide at this moment. We hope that the problem will be solved in the future as new methods of selenophosphate detection become available. We sincerely hope that reviewer accepts this explanation.

Reviewer Figure 3.

(a) SEPHS2 degrades ATP to AMP in the absence of selenium. SEPHS2 U/C was incubated with ATP, and the amount of AMP was assessed by measuring luminescence counts.

(b) Myc-SEPHS2 WT was purified from cells and incubated with SCLY and Sec in the presence of ATP.

Reviewer Figure 4.

(a) ^{31}P NMR spectra derived from the reaction mixture containing SEPHS2. ^1H -decoupled NMR spectra were obtained by JEOL JNM ECP600 (14.1 T). The Larmor frequencies for ^{31}P nuclei at 243 MHz; pulse width: 90 deg; relaxation delay: 5 sec; accumulation: 128 times. (b and c) Production yield of selenophosphate. (b) The reaction mixture contained 100 mM Tricine (pH 7.2), 3 mM MgCl_2 , 2 mM dithiothreitol, 1.5 mM ATP, 1.5 mM NaSeH (generated by reducing Na_2SeO_3 before addition to the reaction mixture), and 10 μM SEPHS2, and 20 μM PRDX6-WT. The mixture was incubated under Ar gas at 37°C . (c) The reaction mixture contained 100 mM Tricine (pH 7.2), 3 mM MgCl_2 , 2 mM dithiothreitol, 1.5 mM ATP, 1.5 mM Sec (generated by reducing $(\text{Sec})_2$ before addition to the reaction mixture), 4 μM SCLY, 10 μM SEPHS2, and 20 μM PRDX6 WT. The mixture was incubated under Ar gas at 37°C .

- Results presented in Figure 4, where the authors feed selenium in both organic and inorganic forms (selenite and selenocysteine), argue against the proposed mechanism since PRDX6 KO cells promptly metabolize these intermediates to generate new selenoproteins and hint at an alternative form. The data does not support the model depicted in Extended Figure 4 where all selenide would travel via PRDX6. While the assays addressing time and concentration-dependent effects are well taken, their interpretation is not straightforward because the PRDX6-KO cells start with an already lower level of expression. An important aspect would be to consider using radiolabeled selenium sources, as this would be the only way to unambiguously identify *de novo* selenoprotein biosynthesis. Additionally, the authors might wish to consider a more physiological source of selenium, for example, SELENOP.

Response: We thank the reviewer for these suggestions. We were also curious about why selenoproteins can be synthesized in PRDX6 KO cells when excess selenium (regardless of organic or inorganic) was added (Fig. 4a). During preparation of the revised manuscript, we unexpectedly found an alternative mechanism. We observed that overexpressed, but not endogenous, SEPHS2 substantially rescued the PRDX6 KO phenotype (Fig. 5a and Extended Data Fig. 6b–f), which implies that SEPHS2 can utilize selenium without the help of PRDX6,

albeit inefficiently. We then modified the schematic presentation of the Sec incorporation mechanism; the new scheme is now included as the new Figure 5j. At the same time, the above results brought to our attention that we could adjust the amount of SEPHS2, or other selenoproteins, in PRDX6 KO cells to levels comparable with those in WT cells by increasing or decreasing the amount of the selenium source in the culture medium. In response to the reviewer's point about differences in selenoprotein expression in WT and PRDX6 KO cells, we adjusted the selenoprotein expression level in WT and PRDX6 KO cells by reducing the concentration of fetal bovine serum, which is the sole source of selenium in our culture medium (Extended Data Fig. 5h), or by preculturing PRDX6 KO cells with organic and inorganic forms of selenium (Extended Data Fig. 5i). We then added selenium to PRDX6 KO and WT cells that express almost equal amounts of selenoproteins (Extended Data Fig. 5i), and found that PRDX6 WT cells are more efficient at utilizing selenium than PRDX6 KO cells. We believe that these results strengthen our conclusion that PRDX6 is involved in efficient selenium utilization. We thank the reviewer for the constructive comment.

Moreover, we have added the SELLENOP data to Figure 4f.

Additional points:

- The authors' validation for the knockouts used in the studies is insufficiently provided, as seen in Figure 1E, F: Supplementary 2c. Given the relevance of this information in determining if the FBXL5-KO dies via ferroptosis in the presence of iron, it is essential to conduct proper characterization.

Response: We thank the reviewer for this suggestion. We have included western blots to validate expression of FBXL5 KO cells (Extended Data Figs. 1f, g and 3a). We have also added data regarding expression of proteins identified by our screening (except LPCAT3; we could not find any antibodies specific for LPCAT3) (Extended Data Figs. 1f, g).

- The quality of some of the western blots, like Figure 3c, needs improvement. Although the authors claim that the reconstitution of PRDX6-WT fully rescues selenoprotein translation, the exposure of GPX1 and GPX4 appears to be extremely low. To ensure a proper comparison, higher exposure should be considered for such experiments.

Response: We thank the reviewer for pointing this out. We have replaced the result shown in the original Figure 3c with a higher exposure image (Fig. 3c).

- Correct lipid peroxides to lipid hydroperoxides.

Response: As suggested by the reviewer, we changed “lipid peroxides” to “lipid hydroperoxides” throughout the manuscript.

- The PLA experiments seem to be done in A549 cells, unlike the rest of the experiments, which are performed in MEF's. The authors need to explain why this experiment was done in another system. The A549 cell line is known to have a mutation in KEAP1 so the "antioxidant" context might be completely different. A validation that loss of PRDX6 in these cells phenocopies the previous results could suffice to put this point to rest.

Response: We intended to do the experiment with MEF cells; however, we were unable to do it due to the poor quality of the antibody. Immunofluorescence staining showed that the anti-PRDX6 antibody that we used also reacted non-specifically with unknown protein(s) in PRDX6 KO MEFs. Since we found that the anti-PRDX6 antibody specifically detected PRDX6 in human cell lines and deletion of PRDX6 from A549 cells greatly reduced expression of selenoproteins (Extended Data Fig. 4d), we utilized A549 cells instead of MEF cells for this experiment. We accept the reviewer's criticism and have conducted the TurboID experiment in MEF cells.

- Fig. 1f. please mark the position of PDSS2 on the volcano plot in Fig. 1d

Response: We have marked the position of PDSS2 in Figure 1d.

Response to Reviewer #2

The paper "PRDX6 augments selenium utilization to limit iron toxicity and ferroptosis" reports peroxiredoxin6 (PRDX6) as a novel component involved in selenoprotein synthesis. Through a genome-wide CRISPR screen for iron-triggered ferroptosis and subsequent gene ontology analysis of the identified ferroptosis suppressors, selenoprotein synthesis factors were identified. Moreover, experimentation on the identified ferroptosis suppressors revealed that the deletion of PRDX6 decreased GPX4 expression. Findings from additional experiments suggested that PRDX6 indirectly suppresses ferroptosis by augmenting GPX4 activity. Because selenoprotein synthesis factors were identified as a top hit from the initial screening and further analysis from database tools suggested a connection between PRDX6 and the selenoprotein synthetic pathway, PRDX6 was hypothesized to be a novel key player involved in selenoprotein synthesis. Further results supported the involvement of PRDX6 in the selenoprotein synthetic pathway and efforts to probe the specific mechanism at play suggested that PRDX6 is a selenide carrier protein that helps facilitate Sec-tRNA[Ser]Sec synthesis. PRDX6 was also evaluated in the context of cancer, in which database analyses revealed higher PRDX6 expression in cancerous tissues compared to normal tissues. In pancreatic cancer cell lines, PRDX6 KO reduced cell viability, while the addition of selenium sources restored viability. Taken together, the identification of PRDX6 as an intracellular selenium carrier involved in the selenoprotein synthesis pathway not only offers novel mechanistic insight, but also suggests the targeting of PRDX6 in this pathway as a potential therapeutic strategy in cancers where high PRDX6 expression is correlated with poor clinical outcomes.

Strengths

- *The systematic nature in which PRDX6 was first identified and then probed, made this paper easy to follow.*
- *The experiments conducted to probe the mechanism by which PRDX6 is involved in the selenoprotein synthetic pathway are thorough and diverse, ranging from CRISPR screening, database analyses, and downstream assays. The in vitro reconstructive system for selenoprotein synthesis developed by the authors was noteworthy.*
- *The figures are high quality.*
- *In addition to dissecting the mechanism in which PRDX6 is involved in selenoprotein synthesis, the authors studied PRDX6 in the context of cancers. This offered more insight into the application of their findings and suggested the targeting of selenoprotein synthesis machinery as a therapeutic strategy in these cancer types.*

Response: We thank the reviewer for thorough evaluation of our manuscript, and for suggesting improvements. We believe that we could address all of the concerns (see our responses below).

*Weaknesses**Major*• *Line 54*

o The authors report that “IRP2 stabilization caused by loss of FBXL5 led to accumulation of redox active iron in cells” Figure 1A suggests that IRP1 may be involved too. Is it possible to say with certainty that IRP2 stabilization led to this? What about IRP1?

Response: As the reviewer points out, IRP1 is also stabilized by loss of FBXL5 because IRP1 lacking Fe-S cluster is also degraded via FBXL5-mediated ubiquitination⁹. However, it is well established that IRP2 is the major IRP that controls iron metabolism in cells under normal conditions. Moreover, loss of IRP2, but not IRP1, restores embryonic lethality in FBXL5 KO mice¹⁰. Thus, we also added the citation of FBXL5 KO mice in text to make sure this issue (lines 87-88).

• *Line 79*

o At this point, it might be too early to say that “the loss of selenoprotein synthesis factors seem to sensitize cells to iron-triggered ferroptosis by reducing GPX4.” This would fit better after bringing up Figure 2C.

Response: We thank the reviewer for this suggestion. We intended to describe the screening results here; however, as the reviewer suggests, the description comes too early in the manuscript. Therefore, we have deleted the phrase “by reducing GPX4” from the sentence (lines 111–113).

• Line 135-136

o The loss of PRDX6 reducing selenoprotein expression across many human cancer cell lines does not suggest that the role of PRDX6 was conserved across species, that would entail including non-human groups as well. Instead, the authors might want to consider re-framing that to suggesting that the role of PRDX6 is conserved across many cancer types.

Response: We thank the reviewer for pointing this out. We have revised the sentence as suggested (line 177).

• Line 227-228

o The authors claim that “inhibition of PRDX6 may provide an effective strategy for sensitizing cancer cells to iron-triggered ferroptosis”
o However, the data suggest that viability is decreased under PRDX6 KO but is rescued with the addition of selenium. Including a ferroptosis inhibitor in these experiments would support the claim that PRDX6 inhibition might sensitize the cancer cells to ferroptosis. As it stands, there is no ferroptosis-specific data in this experiment to make such claim.

Response: We thank the reviewer for suggesting experiments to consolidate our claims. We performed new experiments, including treatment with the ferroptosis inhibitor liproxstatin-1 (Extended Data Fig. 10b,d,h). We believe that our new results clearly show that inhibiting PRDX6 is an effective strategy for sensitizing cancer cells to ferroptosis.

Minor

• Line 37

o Expand GPX4. This is the first time it is mentioned in the main body of the text.

Response: We have defined GPX4 (glutathione peroxidase 4) in the revised manuscript (lines 37–38).

• Line 40

o Expand FBXL5 for the same reason.

Response: We have defined FBXL5 as “F-box and leucine rich repeat protein 5” (lines 70–71).

• Typo in Line 47-line 48:

o “.....revealed (remove extra that) increased peroxidation in iron-repleted FBXL5 KO cells”

Response: We thank the reviewer for pointing this out. We have corrected the sentence (line 79).

• Line 51

o It would be helpful to explicitly discuss how the unbiased genome-wide CRISPR screening was

performed on the FBXL5 KO cells that were treated with FAC or untreated. While this is captured in the figure, it would make it much more clear if explicitly said in the text as well.

Response: We thank the reviewer for this suggestion. We have now described this in the text (lines 83–84).

• *Figure 1B*

o It would be helpful to include the concentrations of liproxstatin in the image itself.

Response: We added the concentration of liproxstatin-1 to Figure 1b.

• *Line 86*

o It might be beneficial to elaborate more on why PRDX6 did not affect iron homeostasis.

Response: We thank the reviewer for pointing this out. We now describe the rationale behind this conclusion (lines 121–124).

• *Line 106*

o Missing the word “a” after “however” and before “controversial”

Response: We thank the reviewer for pointing this out. We found three papers that reported no GPX activity in PRDX6, so we used the plural (line 144)

• *Line 132*

o This is the first time encountering SEPHS2 in the main body of the text- please expand in the first mention.

Response: We have defined SEPHS2 as “Selenophosphate Synthetase 2” (line 47).

• *Line 139*

o Expand PSTK- this is the first mention.

Response: We have defined PSTK as “phosphoseryl-tRNA^{[Ser]Sec} Kinase” (line 181).

• *Line 154*

o Typo. “Finally, we explored the role of PRDX6 in selenoprotein synthesis (or in the selenoprotein synthetic pathway).”

Response: We thank the reviewer for pointing this out. We have corrected the text accordingly (line 188).

- Line 191-193
 - o *This could benefit from some restructuring.*

Response: We have restructured this sentence as suggested (lines 273–274).

- Line 197
 - o *“..and then transfer”*
- Line 198
 - o *“..revealed that endogenous PRDX6 is effectively..”*

Response: We are so sorry, but we could not understand what the reviewer was asking here, so we could not address this point.

Response to Reviewer #3

Remarks to the Author:

In this manuscript, the authors identified PRDX6 as a novel intracellular selenium carrier. They provided impressive evidence to show that PRDX6 augmented the intracellular selenium utilization efficiency as a selenium carrier and promoted selenoprotein synthesis in general. The function of PRDX6 in ferroptosis is somewhat controversial in the previous studies. This study provides a clear mechanism that loss of PRDX6 promotes ferroptosis indirectly decreasing GPX4 abundance. Overall, this study provides novel viewpoint to the field of selenoprotein and ferroptosis. To improve the current version and make the conclusion more convincing, the authors need to conduct additional experiments and address several important points as following:

Response: We thank the reviewer for acknowledging the scientific and biological importance of our study, and for the insightful comments that have strengthened our conclusions. We have conducted additional experiments and believe that have addressed all of the concerns raised. Our point-by-point responses are listed below.

1. The authors performed a secondary screen using the abundance of GPX4 as a readout with sgRNA against a small panel of genes (extended data Fig2C). This screen identified PRDX6 as a negative regulator for GPX4 protein level. However, it is not described how they narrowed down their screen to this list of genes. The authors should provide the rationale here.

Response: We thank the reviewer for pointing this out. We have included the ranking data from the CRISPR screen (see Extended Data Fig. 2c) to show the rationale behind how we narrowed down the screening.

2. PRDX6 is important for GPX4 translation, and its expression is associated with poor

prognosis of several cancers. To improve the significance of this study, the authors can collect a panel of cancer cells (e.g., PDAC) and compare the expression level of PRDX6, GPX4, SEPHS2 in these cell lines, as well as to compare their sensitivity to iron or RSL3-induced ferroptosis.

Response: We thank the reviewer for this suggestion. We have included data showing expression of PRDX6, as well as selenoproteins (including GPX4 and SEPHS2), by several cancer cell lines and compared their sensitivity to iron-triggered ferroptosis (Extended Data Fig. 10e,f). Interestingly, we found that MYC-N amplified neuroblastoma cell lines (SK-N-DZ and NB-1) were highly sensitive to iron-triggered ferroptosis. MYC-N increases iron influx by upregulating expression of the transferrin receptor 1 (TFRC), a protein responsible for iron-uptake¹¹. Thus, iron-induced ferroptosis may be more effective in cancer cells with high iron levels. Moreover, we generated PRDX6 KO cells from SK-N-DZ cells and found that the reduced viability of KO cells was reversed by addition of selenium and liproxstatin-1 (Extended Data Fig. 10h).

3. Unlike other selenoproteins, SEPHS2 is somehow less affected by PRDX6 knockout (fig 3b, 3c, extended Fig 3d, etc.). Knockout of PRDX6 in HepG2 cells even significantly increased SEPHS2 expression. Sine SEPHS2 is responsible for phosphorylating hydroxyselenide to selenophosphate, it is possible that SEPHS2 itself can directly bind to selenide without the help of PRDX6 (although the affinity might be much weaker). The authors tested that SEPHS2 U/C mutant did not rescue PRDX6 KO phenotype, however, they should also test whether SEPHS2 wt overexpression can rescue PRDX6 KO phenotype. In Fig 3D and Fig 4J, they should also test the conditions of PRDX6ko + SEPHS2 wt or SEPHS2 mutant to rule out this possibility.

Response: We thank the reviewer for insightful comment. As the reviewer points out, SEPHS2 is less affected by PRDX6 KO. Since the amount of SEPHS2 was also less affected in cells lacking LRP8¹², which is the receptor for the major selenium source protein SELLENOP, we supposed that expression of SEPHS2 is less affected by reduced selenium availability. Moreover, we observed that excess selenium increased the amount of some selenoproteins in PRDX6 KO cells markedly (Fig. 4a–c). Thus, in response to the reviewer’s comment we tested the hypothesis that SEPHS2 can bind directly to selenide without the help of PRDX6. To do this, we overexpressed SEPHS2 WT in PRDX6 KO cells, and found that overexpression of SEPHS2 WT substantially restored expression of GPX4 (Fig. 5a and Extended Data Fig. 6b), expression of model selenoprotein (Extended Data Fig. 6c), and cell viability (Extended Data Fig. 6d–f). By contrast, overexpression of SEPHS2 in PRDX6 KO cells did not fully restore selenoproteins SELN and GPX1 (Extended Data Fig. 6b), or the amount of Sec-tRNA^{[Ser]Sec} (Fig. 5c). In addition, endogenous expression of SEPHS2 in PRDX6 KO cells failed to restore selenoprotein expression (Fig. 5b). Taken together, these results strongly indicate that SEPHS2 can bind directly to selenide without the help of PRDX6, albeit very inefficiently, and that PRDX6 plays a crucial role in efficient utilization of selenium by SEPHS2. To include these results in the text,

we have reordered the data in the previous Figures 3 and 4. Again, we appreciate the reviewer's insightful comment, which has improved the manuscript.

4. To confirm that PRDX6 suppresses ferroptosis through maintaining GPX4 expression, the authors need to test whether overexpression of PRDX6 wt and mutants (as in Fig2D) can suppress ferroptosis triggered by RSL3 and IKE, in addition to FAC. More importantly, the authors should establish a cellular system with genotype of GPX4-KO/FSP1-overexpression as in PMID 37267948 (these GPX4-null cells can grow normally because of FSP1 overexpression). Based on the mechanism proposed by the authors, in these cells PRDX6 won't be able to suppress IKE-induced ferroptosis.

Response: We thank the reviewer for this constructive comment, which enabled us to strengthen our hypothesis. We performed new experiments evaluating RSL3 and IKE-induced ferroptosis in PRDX6 KO cells expressing WT or mutant PRDX6 (Extended Data Fig. 3e,f). We also established GPX4 KO/FSP1-overexpressing cells, and found that further deletion of PRDX6 did not affect IKE-induced ferroptosis in GPX4 KO/FSP1-overexpressing cells (Fig. 2j). These results indicate clearly that PRDX6 suppresses ferroptosis by maintaining expression of GPX4.

5. The difference of the selenophosphate synthetase assay readout in Fig 4m, 4n is rather marginal. More convincing results are needed here (hopefully this can be achieved by optimizing assay condition?). The SEPHS2-His used in this experiment is purified from bacteria. Guess it is the U-to-C mutant? The authors might need to purify SEPHS2 wt using a eucaryotic expression system.

[N.B. The reply to this point is the same as our reply to point 6 of reviewer #1 but we also include it here for the ease of the reviewer.]

We thank the reviewer for the critical comment. As reviewer points out, we realized that SEPHS2 U/C constitutively degrades ATP to AMP, even in the absence of selenide, as reported previously^{6,7}. Therefore, the increase in SEPHS2 U/C-mediated AMP production upon addition of PRDX6 WT was not large, although it was statistically significant (Fig. 5h, i). To improve the experimental system, we purified SEPHS2 U/C using two tags (GST and MBP) that are different His₆ tag; however, SEPHS2 U/C also degraded ATP to AMP in the absence of selenium, regardless of the tags (Reviewer Fig. 3a). It might be possible that degradation of ATP to AMP in the absence of selenide is lower in SEPHS2 WT than in SEPHS2 U/C. Therefore, we tried to purify SEPHS2 WT. Although a previous report showed that SEPHS2 WT can be purified using a baculovirus expression system, the authors used a construct lacking SECIS within the 3'-UTR⁸. To purify SEPHS2 containing Sec, we generated a recombinant baculovirus containing a SEPHS2 WT construct with a 3'-UTR containing SECIS; however, we could not obtain enough SEPHS2 WT using this system. We then decided to purify SEPHS2 WT from cells overexpressing SEPHS2 WT, although it was poorly expressed (Fig. 5a). Unfortunately, purified

SEPHS2 WT also constitutively degraded ATP to AMP, although addition of selenium increased the generation of AMP (Reviewer Fig. 3b). Finally, we measured selenophosphate synthesis by SEPHS2 directly in NMR experiments. Selenophosphate was detected by NMR using inorganic or organic selenium as a selenium source; however, there was no significant increase in selenophosphate upon addition of PRDX6 (Reviewer Fig. 4a–c). We found that PRDX6 plays a role the efficiency of selenium utilization on a nanomolar scale (Fig. 4d,e and Extended Data Fig. 5f–i). and that SEPHS2 utilizes selenium under conditions of high selenium concentrations, even in the absence of PRDX6 (Fig. 4a–c). We had to add much higher concentrations of selenium (millimolar) to the experiment due to the poor sensitivity of NMR. At such high concentrations, it is likely that selenium would be transferred to SEPHS2 in the absence of PRDX6. Although we tried to improve the experimental procedure to increase the efficacy and detect the selenophosphate synthesis activity of SEPHS2, we were unsuccessful (possibly due to degradation of ATP to AMP by SEPHS2 in the absence of selenium). Thus, we believe that our current results, which show with statistical significance that PRDX6 increases production of AMP by SEPHS2 is the best result that we can provide at this moment. We hope that the problem will be solved in the future as new methods of selenophosphate detection become available. We sincerely hope that reviewer accepts this explanation.

Reviewer Figure 3.

- (a) SEPHS2 degrades ATP to AMP in the absence of selenium. SEPHS2 U/C was incubated with ATP, and the amount of AMP was assessed by measuring luminescence counts.
- (b) Myc-SEPHS2 WT was purified from cells and incubated with SCLY and Sec in the presence of ATP.

Reviewer Figure 4

(a) ^{31}P NMR spectra derived from the reaction mixture containing SEPHS2. ^1H -decoupled NMR spectra were obtained by JEOL JNM ECP600 (14.1 T). The Larmor frequencies for ^{31}P nuclei at 243 MHz; pulse width: 90 deg; relaxation delay: 5 sec; accumulation: 128 times. (b and c) Production yield of selenophosphate. (b) The reaction mixture contained 100 mM Tricine (pH 7.2), 3 mM MgCl_2 , 2 mM dithiothreitol, 1.5 mM ATP, 1.5 mM NaSeH (generated by reducing Na_2SeO_3 before addition to the reaction mixture), and 10 μM SEPHS2, and 20 μM PRDX6-WT. The mixture was incubated under Ar gas at 37°C. (c) The reaction mixture contained 100 mM Tricine (pH 7.2), 3 mM MgCl_2 , 2 mM dithiothreitol, 1.5 mM ATP, 1.5 mM Sec (generated by reducing $(\text{Sec})_2$ before addition to the reaction mixture), 4 μM SCLY, 10 μM SEPHS2, and 20 μM PRDX6 WT. The mixture was incubated under Ar gas at 37°C.

Minor points:

1. In extended data Fig3d, knockout of PRDX6 in HepG2 cells significantly increased SEPHS2 expression but only slightly decreased GPX4 expression. Why is that, and does PRDX6 knockout affect ferroptosis sensitivity in HepG2?

Response: We re-examined expression of SEPHS2 by PRDX6 KO HepG2 cells, and found that expression was indeed increased (Reviewer Fig. 5a). Hepatocytes are important for maintenance of systemic selenium homeostasis; they do this by synthesizing SELENOP. A review of the database revealed that expression of SEPHS2 increased in liver (Reviewer Fig. 5b). Thus, we suspect that hepatocytes may possess a specific mechanism that maintains expression of SEPHS2. As the reviewer hypothesizes, we found that loss of PRDX6 from HepG2 cells had almost no effect on iron-triggered ferroptosis (Reviewer Fig. 5c). Considering that hepatocytes also play a central role in regulating systemic iron metabolism in addition to selenium homeostasis, hepatocytes may have special mechanisms that enable them to evade iron toxicity.

Reviewer Figure 5.

- Immunoblot analysis of lysates from the indicated cells.
- SEPHS2 expression profiles in several types of normal and tumor tissue.
- Viability of the indicated cells treated with FAC.

2. The authors need to show SEPHS2 expression in Fig 4D, 4E.

Response: We have included western blots showing expression of SEPHS2 (Extended Data Fig. 5f, g).

3. Why is PRDX6 not an essential gene in most of the cancer cell lines, which is very different to other core components of selenoprotein translation pathways (as shown by DepMap)?

Response: As mentioned in the reply to major comment #3, SEPHS2 can acquire selenide without the help of PRDX6, albeit inefficiently. Therefore, even in PRDX6 KO cells that express trace amounts of SEPHS2, tiny amounts of selenoproteins (including SEPHS2 and GPX4) can be expressed. Thus, we assume that PRDX6-deficient cells are viable because selenoprotein is not lost completely. We have discussed this point in the “Discussion” section (lines 376-384).

4. Please describe PLA assay in more detail in the method part.

Response: Thank you. We have included a detailed description of the PLA assay in the “Methods” section (lines 537-553).

References

1. Manevich, Y., Feinstein, S.I. & Fisher, A.B. Activation of the antioxidant enzyme 1-CYS peroxiredoxin requires glutathionylation mediated by heterodimerization with pi GST. *Proc. Natl. Acad. Sci. U. S. A.*, **101**, 3780-5 (2004).
2. Liu, G. et al. Comparison of glutathione peroxidase 1 and peroxiredoxin 6 in protection against oxidative stress in the mouse lung. *Free Radic Biol Med* **49**, 1172-81 (2010).
3. Yun, H.M. et al. PRDX6 promotes lung tumor progression via its GPx and iPLA2 activities. *Free Radic. Biol. Med.*, **69**, 367-76 (2014).
4. Chen, C. et al. Identification of peroxiredoxin 6 as a direct target of withangulatin A by quantitative chemical proteomics in non-small cell lung cancer. *Redox Biol.*, **46**, 102130 (2021).
5. Shahnaj, S. et al. Hyperoxidation of Peroxiredoxin 6 Induces Alteration from Dimeric to Oligomeric State. *Antioxidants*, **8** (2019).
6. Veres, Z., Kim, I.Y., Scholz, T.D. & Stadtman, T.C. Selenophosphate synthetase. Enzyme properties and catalytic reaction. *J. Biol. Chem.*, **269**, 10597-603 (1994).
7. Abe, K., Mihara, H., Nishijima, Y., Kurokawa, S. & Esaki, N. Functional Analysis of Two Homologous Mouse Selenophosphate Synthetases. *Biomedical Research on Trace Elements*, **19**, 76-79 (2008)
8. Kim, I.Y., Guimarães, M.J., Zlotnik, A., Bazan, J.F. & Stadtman, T.C. Fetal mouse selenophosphate synthetase 2 (SPS2): characterization of the cysteine mutant form overproduced in a baculovirus-insect cell system. *Proc. Natl. Acad. Sci. U. S. A.*, **94**, 418-21 (1997).
9. Salahudeen, A.A. et al. An E3 ligase possessing an iron-responsive hemerythrin domain is a regulator of iron homeostasis. *Science*, **326**, 722-6 (2009).
10. Moroishi, T., Nishiyama, M., Takeda, Y., Iwai, K. & Nakayama, K.I. The FBXL5-IRP2 axis is integral to control of iron metabolism in vivo. *Cell Metab*, **14**, 339-51 (2011).
11. Lu, Y. et al. MYCN mediates TFRC-dependent ferroptosis and reveals vulnerabilities in neuroblastoma. *Cell Death. Dis.*, **12**, 511 (2021).
12. Li, Z. et al. Ribosome stalling during selenoprotein translation exposes a ferroptosis vulnerability. *Nat. Chem. Biol.*, **18**, 751-761 (2022).

Decision Letter, first revision:

Message: Our ref: NSMB-A47863A

13th Mar 2024

Dear Dr. Iwai,

Thank you for submitting your revised manuscript "PRDX6 augments selenium utilization to limit iron toxicity and ferroptosis" (NSMB-A47863A). It has now been seen by two of the original referees and their comments are below. These experts find that the paper has improved in revision, and we editorially deem that, since the expertise of reviewer #1 who was unavailable during this round of revisions are covered and their major concerns are addressed, no further issues remain. Therefore we are happy to accept the manuscript in principle in Nature Structural & Molecular Biology, pending minor revisions to satisfy the referees' final requests and to comply with our editorial and formatting guidelines.

We are now performing detailed checks on your paper and will send you a checklist detailing our editorial and formatting requirements in about two weeks. Please do not upload the final materials and make any revisions until you receive this additional information from us.

To facilitate our work at this stage, it is important that we have a copy of the main text as a word file. If you could please send along a word version of this file as soon as possible, we would greatly appreciate it; please make sure to copy the NSMB account (cc'ed above).

Sincerely,

Dimitris Typas
Associate Editor
Nature Structural & Molecular Biology
ORCID: 0000-0002-8737-1319

Reviewer #2 (Remarks to the Author):

The authors have done a thorough job in addressing the previous comments of the reviewers, and the manuscript is now suitable for publication.

Reviewer #3 (Remarks to the Author):

The authors have addressed all the comments from this reviewer satisfactorily.

Final Decision Letter:

Message: 7th May 2024

Dear Dr. Iwai,

We are now happy to accept your revised paper "PRDX6 augments selenium utilization to limit iron toxicity and ferroptosis" for publication as an Article in Nature Structural & Molecular Biology.

Over the next few weeks, your paper will be copyedited to ensure that it conforms to

Nature Structural & Molecular Biology style. Once your paper is typeset, you will receive an email with a link to choose the appropriate publishing options for your paper and our Author Services team will be in touch regarding any additional information that may be required.

Your paper will be published online soon after we receive proof corrections and will appear in print in the next available issue. You can find out your date of online publication by contacting the production team shortly after sending your proof corrections.

Please note that *Nature Structural & Molecular Biology* is a Transformative Journal (TJ). Authors may publish their research with us through the traditional subscription access route or make their paper immediately open access through payment of an article-processing charge (APC). Authors will not be required to make a final decision about access to their article until it has been accepted. Find out more about Transformative Journals

Authors may need to take specific actions to achieve compliance with funder and institutional open access mandates. If your research is supported by a funder that requires immediate open access (e.g. according to Plan S principles) then you should select the gold OA route, and we will direct you to the compliant route where possible. For authors selecting the subscription publication route, the journal's standard licensing terms will need to be accepted, including self-archiving policies. Those licensing terms will supersede any other terms that the author or any third party may assert apply to any version of the manuscript.

Sincerely,

Dimitris Typas
Senior Editor
Nature Structural & Molecular Biology
ORCID: 0000-0002-8737-1319